# A Wasserstein-2 Distance for Efficient Reconstruction of Stochastic Differential Equations

## Abstract

We provide an analysis of the squared Wasserstein-2 ($W_2$) distance between two probability distributions associated with two stochastic differential equations (SDEs). Based on this analysis, we propose a novel squared $W_2$ distance-based loss function for efficiently *reconstructing SDEs* from noisy data. To demonstrate the practical use our Wasserstein distance-based loss function, we carry out numerical experiments that show its efficiency in reconstructing SDEs associated with a number of applications.

## 1 Introduction

Stochastic processes are mathematical models of random phenomena that evolve over time or space (Cinlar, 2011). Among stochastic processes, stochastic differential equations (SDE) of the form

$$dX(t) = f(X(t), t)dt + \sigma(X(t), t)dB(t), \ \ t \in [0, T] \tag{1}$$

are widely used to model complex systems with continuous variables and noise in different fields. Here, $f$ and $\sigma$ denote deterministic and stochastic components of the SDE, while $B(t)$ represents the standard Brownian motion. In applications such as computational fluid dynamics, cell biology, and genetics, the underlying dynamics are often unknown or only partially observed, and subjected to noise. Consequently, it is vital to develop methods capable of reconstructing the governing SDEs from available data (Sullivan, 2015; Soize, 2017; Mathelin et al., 2005; Bressloff, 2014; Lin & Buchler, 2018). Traditional methods, such as the Kalman filter (Welch et al., 1995; Welch, 2020) and Gaussian process regression (Liu et al., 2020; MacKay et al., 1998), often assume specific forms of noise. These methods may not be suitable for complex or nonlinear systems where the noise affects the dynamics in a complicated manner.

Recent advancements leverage machine learning, specifically neural ordinary differential equations (NODEs) (Chen et al., 2018), to offer a more flexible approach to reconstructing SDEs in the form of neural SDEs (nSDEs) (Tzen & Raginsky, 2019; Tong et al., 2022; Jia & Benson, 2019). Despite the promise, challenges remain, especially when selecting optimal loss functions (Jia & Benson, 2019). The Wasserstein distance, a family of metrics that measures discrepancies between probability measures over a metric space, has emerged as a potential solution due to its robust properties (Villani et al., 2009; Oh et al., 2019; Zheng et al., 2020). In this paper, we introduce bounds on the second-order Wasserstein $W_2$ distance between two probability distributions over the continuous function space generated by solutions to two SDEs. Our results motivate the use of this distance for SDE reconstruction. We test our approach on different examples to showcase its effectiveness.

### 1.1 Related work

Traditional methods for reconstructing SDEs from data usually make assumptions on the specific forms of the underlying SDE and fit the unknown parameters. For example, (De Vecchi et al., 2016) uses some polynomials to model $f, \sigma$ in Eq. 1, and (Pereira et al., 2010) assumes linear $f$ and $\sigma$ in Eq. 1.

Previous attempts at using neural SDEs (nSDEs) have explored different loss functions for reconstruction. For example, (Tzen & Raginsky, 2019) models the SDE as a continuum limit of latent deep Gaussian models and uses a variational likelihood bound for training. (Kidger et al., 2021) models SDEs and adopts Wasserstein generative adversarial networks (WGANs), proposed in (Arjovsky et al., 2017), for reconstructing SDEs as the generator. (Briol et al., 2019) uses a maximum mean discrepancy (MMD) loss and a generative model for training SDEs. (Song et al., 2020) assumes that $\sigma$ in Eq. 1 depends only on time and uses a score-based generative model for SDE reconstruction.

The Wasserstein distance, denoted as $W$, has gained use in statistics and machine learning. Seminal works have delved into its analysis (Rüschendorf, 1985) and its utilization in reconstructing discrete-time stochastic processes (Bartl et al., 2021). In the context of SDEs, (Bion-Nadal & Talay, 2019) introduced a restricted Wasserstein-type distance, while (Sanz-Serna & Zygalakis, 2021) and (Wang, 2016; Sanz-Serna & Zygalakis, 2021) examined its application in ergodic SDEs, Levy processes, and Langevin equations respectively. Calculating the $W$ distance for multidimensional random variables is challenging; hence, approximations like the sliced $W$ distance and regularized $W$ distance have emerged (Cuturi et al., 2019; Kolouri et al., 2018; 2019; Rowland et al., 2019; Frogner et al., 2015).

The aforementioned WGAN approach in (Kidger et al., 2021) uses the first-order Wasserstein distance to indirectly reconstruct SDEs via the Kantorovich-Rubinstein duality (Arjovsky et al., 2017). To the best of our knowledge, there has been little literature on directly analyzing and applying the $W$ distance for reconstructing SDEs.

## 1.2 OUR CONTRIBUTION

In our paper, we propose and analyze a novel squared-$W_2$-distance SDE reconstruction method. To be more specific, we denote $\mu$ to be the probability distribution over the continuous function space generated by the solution $X(t)$ to Eq. 1. For the following approximation to Eq. 1, where $\hat{B}(t)$ is another standard Brownian motion that is independent of $B(t)$,

$$\mathrm{d}\hat{X}(t) = \hat{f}(X(t), t)\mathrm{d}t + \hat{\sigma}(X(t), t)\mathrm{d}\hat{B}(t), \ t \in [0, T], \tag{2}$$

we denote $\hat{\mu}$ to be the probability distribution over the continuous function space generated by the solution $\hat{X}(t)$ to Eq. 2. To develop our SDE reconstruction method, we

- First, derive an upper bound for the $W_2$ distance $W_2(\mu, \hat{\mu})$ between solutions to two SDEs Eq. 1 and 2 given the same initial condition. To be specific, we establish a $W_2$ distance upper bound which depends explicitly on the difference in the drift and diffusion functions $f - \hat{f}$ and $\sigma - \hat{\sigma}$ associated with using Eq. 2 to approximate the SDE in Eq. 1. This bound is developed in Theorem 1 of Section 2.

- Next, we prove that the $W_2$ distance of two SDEs $W_2(\mu, \hat{\mu})$ can be approximated by estimating the $W_2$ distance between their finite-dimensional projections. This result helps us define a simple squared-$W_2$-distance loss function for reconstructing SDEs, as detailed in Theorem 2 of Section 2.

- Finally, in Section 3, we carry out numerical experiments and find that our squared-$W_2$-distance loss function performs better than other loss functions commonly used for uncertainty quantification across many SDE reconstruction problems. Furthermore, we propose ways to generalize our $W_2$-distance method to the reconstruction of multidimensional SDEs.

## 2 SQUARED $W_2$ DISTANCE FOR RECONSTRUCTING SDES

In this section, we prove the bounds for the squared $W_2$ distance of two probability measures associated with two SDEs. Specifically, we demonstrate that minimizing the squared $W_2$ distance leads to efficient

reconstruction of $f, \sigma$ in Eq. 1. We also show how to estimate the squared $W_2$ between two probability measures associated with two SDEs by using their finite dimensional projections. Based on our analysis, we propose a simple squared $W_2$ distance loss function Eq. 17 for reconstructing SDEs. If we observe the $X(t_i)$ obeying a one-dimensional SDE at uniformly spaced time points $t_i = \frac{iT}{N}, \ i = 0, ..., N$, our proposed loss function is simply

$$\Delta t \sum_{i=1}^{N-1} \int_0^1 (F_i^{-1}(s) - \hat{F}_i^{-1}(s))^2 \mathrm{d}s, \tag{3}$$

where $\Delta t$ is the timestep and $F_i$ and $\hat{F}_i$ are the empirical cumulative distribution functions for $X(t_i)$ and $\hat{X}(t_i)$, respectively.

First, we follow the definition of the squared $W_2$ distance in (Clement & Desch, 2008) for two probability measures $\mu, \hat{\mu}$ associated with two continuous stochastic processes $X(t), \hat{X}(t), \ t \in [0, T]$.

**Definition 2.1.** For two $d$-dimensional continuous stochastic processes in the separable space $\big(C([0, T]; \mathbb{R}^d), \|\cdot\|\big)$

$$\boldsymbol{X}(t) = (X^1(t), ..., X^d(t)) \in C([0, T]; \mathbb{R}^d), \ \hat{\boldsymbol{X}}(t) = (\hat{X}^1(t), ..., \hat{X}^d(t)) \in C([0, T]; \mathbb{R}^d), \ t \in [0, T], \tag{4}$$

with two associated probability distributions $\mu, \hat{\mu}$, the squared $W_2(\mu, \hat{\mu})$ distance between $\mu, \hat{\mu}$ is defined as

$$W_2^2(\mu, \hat{\mu}) = \inf_{\pi(\mu, \hat{\mu})} \mathbb{E}_{(\boldsymbol{X}, \hat{\boldsymbol{X}}) \sim \pi(\mu, \hat{\mu})}\big[\|\boldsymbol{X} - \hat{\boldsymbol{X}}\|^2\big], \tag{5}$$

where the distance $\|\cdot\|$ is defined as $\|\boldsymbol{X}\| := \big(\int_0^T \sum_{i=1}^d |X_i(t)|^2 \mathrm{d}t\big)^{\frac{1}{2}}$ and $\pi(\mu, \hat{\mu})$ iterates over all *coupled* distributions of $\boldsymbol{X}(t), \hat{\boldsymbol{X}}(t)$, defined by the condition

$$\begin{cases} \boldsymbol{P}_{\pi(\mu, \hat{\mu})}\big(A \times C([0, T]; \mathbb{R}^d)\big) = \boldsymbol{P}_\mu(A), \\ \boldsymbol{P}_{\pi(\mu, \hat{\mu})}\big(C([0, T]; \mathbb{R}^d) \times A\big) = \boldsymbol{P}_{\hat{\mu}}(A), \end{cases} \forall A \in \mathcal{B}\big(C([0, T]; \mathbb{R}^d)\big), \tag{6}$$

where $\mathcal{B}\big(C([0, T]; \mathbb{R}^d)\big)$ denotes the Borel $\sigma$-algebra associated with the space of $d$-dimensional continuous functions $C([0, T]; \mathbb{R}^d)$.

We shall first prove an upper bound for the $W_2$ distance between the probability measures $\mu$ and $\hat{\mu}$ associated with $X(t), \hat{X}(t)$, solutions to Eq. 1 and Eq. 2, respectively.

**Theorem 1.** If $\{X(t)\}_{t=0}^T, \{\hat{X}(t)\}_{t=0}^T$ have the same initial condition distribution and they are solutions to Eq. 1 and Eq. 2 in the univariate case ($d = 1$ in Eq. 4), respectively, and the following conditions hold:

- $f, \hat{f}, \sigma, \hat{\sigma}$ are continuously differentiable; $\partial_x \sigma$ and $\partial_x \hat{\sigma}$ are uniformly bounded

- there exists two functions $\eta_1(x_1, x_2), \eta_2(x_1, x_2)$ such that their values are in $(x_1, x_2)$ and $f(X_1, t) - f(X_2, t) = \partial_x f(\eta_1(X_1, X_2), t)(X_1 - X_2)$ and $\sigma(X_1, t) - \sigma(X_2, t) = \partial_x \sigma(\eta_2(X_1, X_2), t)(X_1 - X_2)$

then

$$W_2^2(\mu, \hat{\mu}) \leq 3 \int_0^T \mathbb{E}\Big[\int_0^t e^{2\int_s^t h(X(r), \tilde{X}(r), r)\mathrm{d}r + 2\int_s^t \partial_x \sigma(\eta_2(X(r) - \tilde{X}(r)), r)\mathrm{d}B(r)}\mathrm{d}s\Big]\mathrm{d}t$$

$$\times \mathbb{E}\Big[\int_0^T (f - \hat{f})^2(\tilde{X}(t), t)\mathrm{d}t\Big] + 3 \int_0^T \mathbb{E}\Big[\int_0^t e^{2\int_s^t h(X(r), \tilde{X}(r), r)\mathrm{d}r + 2\int_s^t \partial_x \sigma(\eta_2(X(r), \tilde{X}(r)), r)\mathrm{d}B(r)}\mathrm{d}s\Big]\mathrm{d}t$$

$$\times \mathbb{E}\Big[\int_0^T \big(\partial_x \sigma(\eta_2(X(t), \tilde{X}(t)), t)\big)^2 \cdot (\sigma - \hat{\sigma})^2(\tilde{X}(t), t)\mathrm{d}t\Big]$$

$$+ 3 \int_0^T \mathbb{E}\Big[\int_0^t e^{4\int_s^t h(X(r), \tilde{X}(r), r)\mathrm{d}r + 4\int_s^t \partial_x \sigma(\eta_2(X(r), \tilde{X}(r)), r)\mathrm{d}B(r)}\mathrm{d}s\Big]^{\frac{1}{2}}\mathrm{d}t \times \mathbb{E}\Big[\int_0^T (\sigma - \hat{\sigma})^4(\tilde{X}(t), t)\mathrm{d}t\Big]^{\frac{1}{2}}, \tag{7}$$

where $\tilde{X}(t)$ satisfies

$$\mathrm{d}\tilde{X}(t) = \hat{f}(\tilde{X}(t), t)\mathrm{d}t + \hat{\sigma}(\tilde{X}(t), t)\mathrm{d}B(t), \ \tilde{X}(0) = X(0), \tag{8}$$

and $h$ is defined as

$$h(X(r), \hat{X}(r), r) := \partial_x f\big(\eta_1(X(r), \tilde{X}(r)), r\big) - \frac{1}{2}\Big(\partial_x \sigma\big(\eta_2(X(r), \tilde{X}(r)), r\big)\Big)^2. \tag{9}$$

The proof of Theorem 1 is given in Appendix A. Theorem 1 indicates that as long as

$$\mathbb{E}\Big[\int_0^t e^{4\int_s^t h(X(r), \tilde{X}(r), r)\mathrm{d}r + 4\int_s^t \partial_x \sigma(\eta_2(X(r), \tilde{X}(r)), r)\mathrm{d}B(r)}\mathrm{d}s\Big] \tag{10}$$

is uniformly bounded, the upper bound for $W_2(\mu, \hat{\mu}) \to 0$ when $\hat{f} - f \to 0$ and $\hat{\sigma} - \sigma \to 0$ uniformly in $\mathbb{R} \times [0, T]$. Specifically, if $f = \hat{f}, \sigma = \hat{\sigma}$, then the RHS Eq. 7 is 0. This indicates that minimizing $W_2(\mu, \hat{\mu})$ is necessary for generating small errors $\hat{f} - f, \hat{\sigma} - \sigma$ in order to accurately approximate both $f$ and $\sigma$. MSE-based loss functions (defined in Appendix D) suppress noise. On the other hand, using KL divergence does not ensure a finite divergence between $X(t)$ and $\hat{X}(t)$ even if $\hat{f}$ approximates $f$ and $\hat{\sigma}$ approximates $\sigma$. Detailed discussions on the limitations of MSE and KL divergence in SDE reconstruction can be found in Appendix B.

**Remark:** Theorem 1 may be generalized to some higher dimensional cases when $\boldsymbol{X}(t) := (X^1(t), ..., X^d(t)), \hat{\boldsymbol{X}}(t) := (\hat{X}^1(t), ..., \hat{X}^d(t))$ could be described by

$$\mathrm{d}\boldsymbol{X}(t) = \boldsymbol{f}(\boldsymbol{X}(t), t)\mathrm{d}t + \boldsymbol{\sigma}(\boldsymbol{X}(t), t)\mathrm{d}\boldsymbol{B}(t), \ \mathrm{d}\hat{\boldsymbol{X}}(t) = \hat{\boldsymbol{f}}(\hat{\boldsymbol{X}}(t), t)\mathrm{d}t + \hat{\boldsymbol{\sigma}}(\hat{\boldsymbol{X}}(t), t)\mathrm{d}\hat{\boldsymbol{B}}(t) \tag{11}$$

where $\boldsymbol{f}, \hat{\boldsymbol{f}} : \mathbb{R}^{d+1} \to \mathbb{R}^d$ are the $d$-dimensional drift functions and $\boldsymbol{\sigma}, \hat{\boldsymbol{\sigma}} : \mathbb{R}^{d+1} \to \mathbb{R}^{d \times s}$ are diffusion matrices. $\boldsymbol{B}(t)$ and $\hat{\boldsymbol{B}}(t)$ are two independent $s$-dimensional standard Brownian motions. Under some additional assumptions, one can derive an upper bound for the $W_2$ distance of the probability distributions for $\boldsymbol{X}, \hat{\boldsymbol{X}}$ as in Eq. 7. For example, if for every $i = 1, ...d$, the $i^{\text{th}}$ component $\mathrm{d}X_i(t) = f_i(X_i(t), t)\mathrm{d}t + \sigma^i(X_i(t), t)\mathrm{d}B_i(t)$ and $\mathrm{d}\hat{X}_i(t) = \hat{f}_i(\hat{X}_i(t), t)\mathrm{d}t + \hat{\sigma}_i(\hat{X}_i(t), t)\mathrm{d}\hat{B}_i(t)$, where $B_i(t), \hat{B}_i(t)$ are independent Brownian motions, then similar conclusions could be derived by calculating the difference $X_i - \hat{X}_i$. Developing an upper bound to the $W_2$ distance for general dimensions $d$ requires additional assumptions to find expressions for $\boldsymbol{X} - \hat{\boldsymbol{X}}$. We leave this nontrivial derivation as future work. Although without a formal theoretical analysis, we propose in Example 3.4 a rotated squared-$W_2$-distance loss function that will be shown to be effective in reconstructing multidimensional SDEs.

Our next theorem provides a way to estimate the $W_2$ distance by using finite dimensional projections. In general, we only have finite observations of trajectories for $\{\boldsymbol{X}(t)\}$ and $\{\hat{\boldsymbol{X}}(t)\}$ at discrete time points. Our next result provides an estimate of the $W_2$ between of the probability measures $\mu, \hat{\mu}$ associated with $\boldsymbol{X}(t)$ and $\hat{\boldsymbol{X}}(t)$, $t \in [0, T]$ using their finite-dimensional projections. We assume that $\boldsymbol{X}(t), \hat{\boldsymbol{X}}(t)$ solve the two SDEs described by Eq. 11. We let $0 = t_0 < t_1 < ... < t_N = T, t_i = i\Delta t, \Delta t := \frac{T}{N}$ be a uniform mesh in time, and we define the following projection operator $\boldsymbol{I}_N$

$$\boldsymbol{X}_N(t) := \boldsymbol{I}_N \boldsymbol{X}(t) = \begin{cases} \boldsymbol{X}(t_i), t \in [t_i, t_{i+1}), i < N - 1, \\ \boldsymbol{X}(t_i), t \in [t_i, t_{i+1}], i = N - 1. \end{cases} \tag{12}$$

As in the previous case, we require $\boldsymbol{X}(t)$ and $\hat{\boldsymbol{X}}(t)$ to be continuous. Note that the projected process is no longer continuous, we define a new space $\tilde{\Omega}_N$ containing all continuous and piecewise constant functions and $\mu, \hat{\mu}$ are allowed to be defined on $\tilde{\Omega}_N$ naturally, see Appendix C for details. Distributions over $\tilde{\Omega}_N$ generated by $\boldsymbol{X}_N(t), \hat{\boldsymbol{X}}_N(t)$ in Eq. 12 is denoted by $\mu_N$ and $\hat{\mu}_N$, respectively. We can then estimate $W_2(\mu, \hat{\mu})$ by $W_2(\mu_N, \hat{\mu}_N)$ using the following theorem.

**Theorem 2.** Suppose $\{\boldsymbol{X}(t)\}_{t=0}^T$ and $\{\hat{\boldsymbol{X}}(t)\}_{t=0}^T$ are both continuous-time continuous-space stochastic processes in $\mathbb{R}^d$ and $\mu, \hat{\mu}$ are their associated probability measures, then $W_2(\mu, \hat{\mu})$ can be bounded by their finite-dimensional projections

$$W_2(\mu_N, \hat{\mu}_N) - W_2(\mu, \mu_N) - W_2(\hat{\mu}, \hat{\mu}_N) \le W_2(\mu, \hat{\mu}) \le W_2(\mu_N, \hat{\mu}_N) + W_2(\mu, \mu_N) + W_2(\hat{\mu}, \hat{\mu}_N) \quad (13)$$

where $\mu_N, \hat{\mu}_N$ are the probability distributions associated with $\boldsymbol{X}_N$ and $\hat{\boldsymbol{X}}_N$ defined in Eq. 12. Specifically, if $\boldsymbol{X}(t)$ and $\hat{\boldsymbol{X}}(t)$ solve Eq. 11, and if

$$
\begin{aligned}
F &:= \mathbb{E}\Big[\int_0^T \sum_{i=1}^d f_i^2(\boldsymbol{X}(t), t)\mathrm{d}t\Big] < \infty, \ \Sigma := \mathbb{E}\Big[\int_0^T \sum_{\ell=1}^d \sum_{j=1}^s \sigma_{i,j}^2(\boldsymbol{X}(t), t)\mathrm{d}t\Big] < \infty, \\
\hat{F} &:= \mathbb{E}\Big[\int_0^T \sum_{i=1}^d \hat{f}_i^2(\hat{\boldsymbol{X}}(t), t)\mathrm{d}t\Big] < \infty, \ \hat{\Sigma} := \mathbb{E}\Big[\int_0^T \sum_{\ell=1}^d \sum_{j=1}^s \hat{\sigma}_{i,j}^2(\hat{\boldsymbol{X}}(t), t)\mathrm{d}t\Big] < \infty,
\end{aligned}
\quad (14)
$$

hold, then we obtain the following bound

$$
\begin{aligned}
W_2(\mu_N, \hat{\mu}_N) - \sqrt{(s+1)\Delta t}\left((F\Delta t + \Sigma)^{\frac{1}{2}} + (\hat{F}\Delta t + \hat{\Sigma})^{\frac{1}{2}}\right) &\le W_2(\mu, \hat{\mu}) \\
&\le W_2(\mu_N, \hat{\mu}_N) + \sqrt{(s+1)\Delta t}\left((F\Delta t + \Sigma)^{\frac{1}{2}} + (\hat{F}\Delta t + \hat{\Sigma})^{\frac{1}{2}}\right).
\end{aligned}
\quad (15)
$$

The proof of Theorem 2 is provided in Appendix C. Equation 13 is the triangular inequality for metrics, while Eq. 15 arises from the use of a specific coupling and the Cauchy inequality to bound $W_2(\mu, \mu_N)$ and $W_2(\hat{\mu}, \hat{\mu}_N)$. Theorem 2 gives bounds for approximating the $W_2$ distance between $\boldsymbol{X}(t), \hat{\boldsymbol{X}}(t)$ w.r.t. to their finite dimensional projections $\boldsymbol{X}_N(t), \hat{\boldsymbol{X}}_N(t)$. Specifically, if $\boldsymbol{X}(t), \hat{\boldsymbol{X}}(t)$ are solutions to Eq. 1 and Eq. 2, then as the timestep $\Delta t \to 0$, $W_2(\mu_N, \hat{\mu}_N) \to W_2(\mu, \hat{\mu})$. Theorem 2 indicates that we can use $W_2^2(\mu_N, \hat{\mu}_N)$, which approximates $W_2^2(\mu, \hat{\mu})$ when $\Delta t \to 0$, as a loss function. Furthermore,

$$W_2^2(\mu_N, \hat{\mu}_N) = \inf_{\pi(\mu_N, \hat{\mu}_N)} \sum_{i=1}^{N-1} \mathbb{E}_{(\boldsymbol{X}_N, \hat{\boldsymbol{X}}_N) \sim \pi(\mu_N, \hat{\mu}_N)}\Big[\big|\boldsymbol{X}(t_i) - \hat{\boldsymbol{X}}(t_i)\big|_2^2\Big]\Delta t. \quad (16)$$

Here, $\pi(\mu_N, \hat{\mu}_N)$ iterates over coupled distributions of $\boldsymbol{X}_N(t), \hat{\boldsymbol{X}}_N(t)$, whose marginal distributions coincide with $\mu_N$ and $\hat{\mu}_N$. $|\cdot|_2$ denotes the $\ell^2$ norm of a vector. Note that $\mu_N$ is fully characterized by values of $\boldsymbol{X}(t)$ at the discrete time points $t_i$. Alternatively, we can disregard the temporal correlations of values at different times and relax the constraint on the coupling $\pi(\mu_N, \hat{\mu}_N)$ in Eq. 16, as in (Chewi et al., 2021). We minimize individual terms in the sum with respect to the coupling $\pi_i$ of $\boldsymbol{X}(t_i)$ and $\hat{\boldsymbol{X}}(t_i)$ and define a heuristic loss function

$$\sum_{i=1}^{N-1} \inf_{\pi_i} \mathbb{E}_{\pi_i}\big[|\boldsymbol{X}(t_i) - \hat{\boldsymbol{X}}(t_i)|_2^2\big]\Delta t \le W_2^2(\mu_N, \hat{\mu}_N). \quad (17)$$

Eq. 17 is a lower bound of Eq. 16. For 1D SDEs, we will show convergence of LHS in Eq. 17 to 0 is necessary to have $f - \hat{f}, \sigma - \hat{\sigma} \to 0$ as $N \to \infty$ in Appendix J under certain conditions. In Example 3.4 and Appendix J, we shall also compare the performance of the two losses defined by Eqs. 16 and 17.

## 3 NUMERICAL EXPERIMENTS

We carry out experiments to investigate the efficiency of our proposed squared-$W_2$ loss function (Eq. 17) by comparing it to other methods and loss functions. Our code is available in the supplemental material. Our approach is tested on the reconstruction of several representative SDEs in Examples 3.1, 3.2, and 3.3. In Example 3.4, we propose a way to generalize Eq. 17 to reconstruct a 2D SDE.

In all experiments, we use two neural networks to parameterize $\hat{f} := \hat{f}(X, t; \Theta_1), \hat{\sigma} := \hat{\sigma}(X, t; \Theta_2)$ in Eq. 2 for the purpose of reconstructing $f, \sigma$ in Eq. 1 by the estimates $\hat{f} \approx f, \hat{\sigma} \approx \sigma$. $\Theta_1, \Theta_2$ are the parameter sets in the two neural networks for parameterizing $\hat{f} = \hat{f}_{\Theta_1}, \hat{\sigma} = \hat{\sigma}_{\Theta_2}$. We use the `sdeint` function in the `torchsde` Python package (Li et al., 2020) for numerically integrating SDEs. Details of the training steps for all examples are given in Appendix E.

First, we compare our proposed squared-$W_2$-distance loss (Eq. 17) with several traditional statistical methods for SDE reconstruction.

**Example 3.1.** We reconstruct a nonlinear SDE of the form

$$\mathrm{d}X(t) = \left(\tfrac{1}{2} - \cos X(t)\right)\mathrm{d}t + \sigma \mathrm{d}B(t), \quad t \in [0, 20], \tag{18}$$

which defines a Brownian process in a potential of the form $U(x) = \frac{x}{2} - \sin x$. In the absence of noise, there are infinitely many stable equilibrium points $x_k = \frac{5\pi}{3} + 2\pi k, k \in \mathbb{Z}$. When noise $\sigma \mathrm{d}B(t)$ is added, trajectories tend to saturate around those equilibrium points but jumping from one equilibrium point to another is possible. In this example, we wish to reconstruct $f(x) = \frac{1}{2} - \cos x, \sigma(x) = \frac{1}{2}$. We use the MSE, the Mean²+Variance, the maximum-log-likelihood, and the proposed squared $W_2$ distance Eq. 17 as loss functions. For all loss functions, we use the same neural network hyperparameters. Definitions of all loss functions and training details are provided in Appendix D. Neural networks with the same hidden layers and neurons in each layer are used for each loss function, as detailed in Appendix E. Using the initial condition $X(0) = 0$, the sampled ground-truth and reconstructed trajectories are shown in Fig. 1.

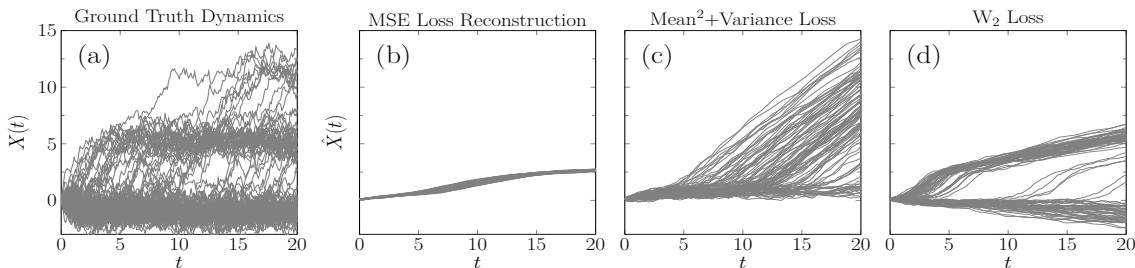

Figure 1: (a) Ground-truth trajectories. (b) Reconstructed trajectories from nSDE using MSE loss. (c) Reconstructed trajectories from nSDE using Mean²+Variance loss. (d) Reconstructed trajectories from nSDE using $W_2$ loss. The max-log-likelihood gives almost the same straight lines for all realizations, which is the worst approximation and not shown.

Fig. 1(a) shows the distributions of 100 trajectories with most of them concentrated around two attractors (local minima $x = -\frac{\pi}{3}, \frac{5\pi}{3}$ of the potential $U(x)$). Fig. 1(b) shows that using MSE gives almost deterministic trajectories that fails to reconstruct the noise. From 1 (c), the Mean²+Variance loss fails to reconstruct the two local equilibrium because the mean²+Variance loss cannot inform on the shape of the distribution of the trajectories at any fixed timepoint. Fig. 1(d) shows that when using our proposed squared $W_2$ loss Eq. 17, the trajectories of the reconstructed SDE can successfully learn the two-attractor feature and potentially the distribution of trajectories. The reason why the reconstructed trajectories of the $W_2$ distance cannot recover the third stable equilibrium at $x = \frac{11\pi}{3}$ is because the data is sparse near it.

In the next example, we show that using our squared $W_2$ distance loss function Eq. 17 leads to efficient reconstruction of $f$ and $\sigma$. We shall use the following mean relative $L^2$ errors in the reconstructed $\hat{f}, \hat{\sigma}$ in Eq. 2 versus the ground-truth $f$ and $\sigma$ in Eq. 1 for evaluating the SDE reconstruction accuracy

$$\left(\sum_{i=0}^{T} \frac{\sum_{j=1}^{N} \|f(x_j(t_i), t_i) - \hat{f}(x_j(t_i), t_i)\|^2}{(T+1)\sum_{j=1}^{N} \|f(x_j(t_i), t_i)\|^2}\right)^{\frac{1}{2}}, \quad \left(\sum_{i=0}^{T} \frac{\sum_{j=1}^{N} \||\sigma(x_j(t_i), t_i)| - |\hat{\sigma}(x_j(t_i), t_i)|\|^2}{(T+1)\sum_{j=1}^{N} \|\sigma(x_j(t_i), t_i)\|^2}\right)^{\frac{1}{2}} \tag{19}$$

where $x_j(t_i)$ is the value of the $j^{\text{th}}$ ground-truth trajectory at $t_i$.

**Example 3.2.** Next, we reconstruct a Cox-Ingersoll-Ross (CIR) model which is commonly used in finance for describing the evolution of interest rates:

$$\mathrm{d}X(t) = \big(5 - X(t)\big)\mathrm{d}t + \sigma_0\sqrt{X(t)}\mathrm{d}B(t), \ \ t \in [0, 2]. \tag{20}$$

Specifically, we are interested in how our reconstructed $\hat{f}, \hat{\sigma}$ can approximate the ground-truth $f(X) = 5 - X$ and $\sigma(X) = \sigma_0\sqrt{X}$, with $\sigma_0$ a parameter. Here, we take the timestep $\Delta t = 0.05$ in Eq. 17 and the initial condition is taken to be $X(0) = 2$. For reconstructing $f, \sigma$, we compare using our proposed squared $W_2$ distance Eq. 17 with those derived by minimizing a Maximum Mean Discrepancy (MMD) loss (Briol et al., 2019) and by using the other loss functions given in Appendix D. Hyperparameters in the neural networks and used for training are the same across all loss functions.

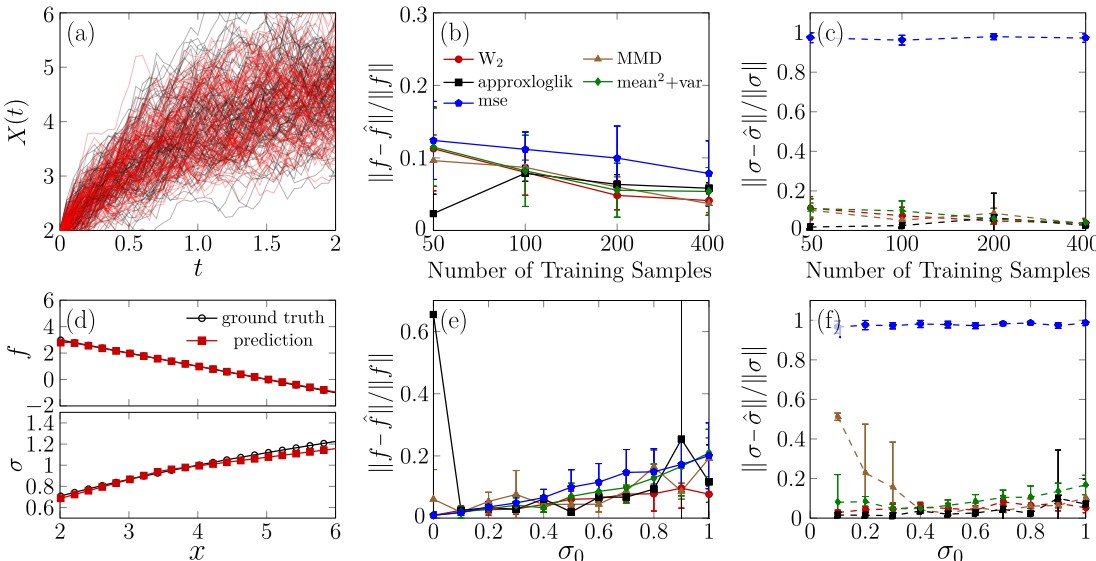

Figure 2: (a) Ground-truth trajectories and reconstructed trajectories by nSDE using squared $W_2$ loss with $\sigma_0 = 0.5$. (b-c) Errors with respect to the numbers of ground-truth trajectories with $\sigma_0 = 0.5$. (d) Comparison of the reconstructed $\hat{f}_{\Theta_1}(u), \hat{\sigma}_{\Theta_2}(u)$ to the ground-truth functions $f(u), \sigma(u)$ with $\sigma_0 = 0.5$. (e-f) Errors with respect to noise level $\sigma_0$ with 200 training samples. Legends for panels (c, e, f) are the same as the one in (b).

Fig. 2(a) shows the predicted trajectories using our proposed squared $W_2$ loss function match well with the ground-truth trajectories. Fig. 2(b, c) indicate that our proposed squared $W_2$ distance loss yields smaller errors defined in Eq. 19 in $f, \sigma$ than other methods when the number of ground-truth trajectories is larger than $\sim 100$. More specifically, we plot the reconstructed $\hat{f}_\Theta, \hat{\sigma}_\Theta$ by using our squared $W_2$ loss in Fig. 2(d); these reconstructions also match well with the ground-truth values $f, \sigma$. When we vary $\sigma_0$ in Eq. 20, our proposed $W_2$ loss function gives the best performance among all loss functions shown in Fig. 2(e, f).

Next, we reconstruct the Ornstein-Uhlenbeck (OU) process given in (Kidger et al., 2021) to compare our loss function with the WGAN-SDE method therein and with another recent MMD method.

**Example 3.3.** We consider reconstructing the following time-inhomogeneous OU process

$$\mathrm{d}X(t) = \big(0.02t - 0.1X(t)\big)\mathrm{d}t + 0.4\mathrm{d}B(t), \ \ t \in [0, 63]. \tag{21}$$

We take the timestep $\Delta t = 1$ in Eq. 17 and the initial condition is taken as $X(0) = 0$. We compare the numerical performance of minimizing Eq. 17 with the WGAN method and using the MMD loss metric. Neural networks with the same number of hidden layers and neurons in each layer are used for all three methods (see Appendix E).

Specifically, in addition to the relative error in the reconstructed $\hat{f}, \hat{\sigma}$, we shall also compare the runtime and memory usage for the three methods when giving different numbers of ground-truth trajectories for training.

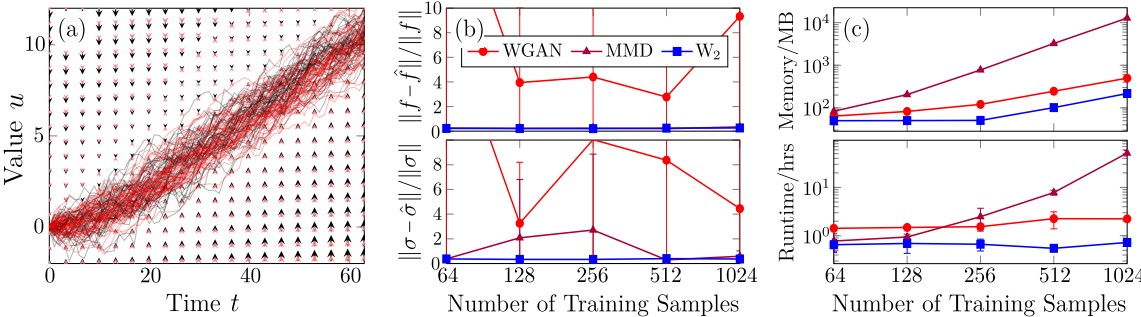

Figure 3: (a) Ground-truth and reconstructed trajectories using the squared $W_2$ loss Eq. 17. Black curves are the ground-truth, red curves are reconstructed trajectories. Black and red arrows indicate $f(x, t)$ and the reconstructed $\hat{f}(x, t)$ at fixed $(x, t)$, respectively. (b) Relative errors in reconstructed $\hat{f}$ and $\hat{\sigma}$, repeated 10 times. Error bars show the standard deviation. (c) Resource consumption with respect to $N_{\text{samples}}$. Memory consumption is measured by torch profiler and represents peak memory usage during training.

From Fig. 3(a), the distribution of trajectories of the reconstructed SDE found from using our proposed squared-$W_2$ loss matches well with the distribution of the ground-truth trajectories. Our method outperforms the other two methods in the relative $L^2$ error of the reconstructed $f, \sigma$ for all different numbers of ground-truth trajectories. If $N_{\text{sample}}$ is the number of training samples, the memory cost in using Eq. 17 is $O(N_{\text{sample}})$; however, the number of operations needed is $O(N_{\text{sample}} \log N_{\text{sample}})$ because we need to reorder the ground-truth $X(t_i)$ and predicted $\hat{X}(t_i)$ data to obtain the empirical cumulative distributions at every $t_i$. On the other hand, the MMD method needs to create an $N_{\text{sample}} \times N_{\text{sample}}$ matrix and thus its memory cost and operations needed are at best $O(N_{\text{sample}}^2)$. The WGAN-SDE method needs to create a generator and a discriminator and its training is complex, leading to both a higher memory cost and a larger runtime than our method. For reconstructing SDEs, a larger number of ground-truth trajectories leads to higher accuracy (see Appendix H). Even if we consider using the SGD to minibatch for training, we find that the batch size cannot be set too small due to the intrinsic noisy nature of trajectories of the SDE to be reconstructed; thus using our squared $W_2$ distance could be overall more efficient than using the MMD method. Additional results are given in Appendix H.

Finally, we carry out an experiment on reconstructing a 2D correlated geometric Brownian motion. We shall propose an extension of the 1D squared $W_2$ distance loss Eq. 17 for reconstructing 2D SDEs.

**Example 3.4.** We consider reconstructing the following 2D correlated geometric Brownian motion that could represent prices of two correlated stocks (Musiela & Rutkowski, 2006)

$$\mathrm{d}X_1(t) = \mu_1 X_1(t) + \sum_{i=1}^{2} \sigma_{1,i} X_i(t)\mathrm{d}B_i(t), \quad \mathrm{d}X_2(t) = \mu_2 X_2(t) + \sum_{i=1}^{2} \sigma_{2,i} X_i(t)\mathrm{d}B_i(t), \ t \in [0, 2], \ (22)$$

where $B_1(t)$ and $B_2(t)$ are independent Brownian processes. In this example, $\boldsymbol{f} := (\mu_1 X_1, \mu_2 X_2)$ is a 2D vector and $\boldsymbol{\sigma} := [\sigma_{1,1} X_1, \sigma_{1,2} X_2; \sigma_{2,1} X_1, \sigma_{2,2} X_2]$ is a $2 \times 2$ matrix. We take $(\mu_1, \mu_2) = (0.1, 0.2)$, and $\boldsymbol{\sigma} =$

$[0.2X_1, -0.1X_2; -0.1X_1, 0.1X_2]$. The initial condition is set to be $(X_1(0), X_2(0)) = (1, 0.5)$. In addition to directly minimizing a 2D version of the squared $W_2$ distance Eq. 17, we consider minimizing a sliced squared $W_2$ distance as proposed in (Kolouri et al., 2018; 2019). To resolve the correlation between $X_1, X_2$, we propose a rotated squared $W_2$ distance. Finally, we also numerically estimate the $W_2$ distance Eq. 16 as well as the time-decoupled approximation Eq. 17 using the `ot.emd2` function in the Python Optimal Transport package Flamary et al. (2021). Formulas of the above five loss functions are in Appendix D. Hyperparameters in the neural networks and for training are the same for using different loss functions. Note that since the SDE has two components, the definition of the relative error in $\sigma$ is revised to

$$\left[ \sum_{i=0}^{T} \frac{\sum_{j=1}^{N} \|\boldsymbol{\sigma}\boldsymbol{\sigma}^T(x_j(t_i), t_i) - \hat{\boldsymbol{\sigma}}\hat{\boldsymbol{\sigma}}^T(x_j(t_i), t_i)\|_F^2}{(T+1)\sum_{\ell=1}^{N} \|\hat{\boldsymbol{\sigma}}\hat{\boldsymbol{\sigma}}^T(x_\ell(t_i), t_i)\|_F^2} \right]^{1/2}, \tag{23}$$

where $\|\cdot\|_F$ is the Frobenius norm for matrices.

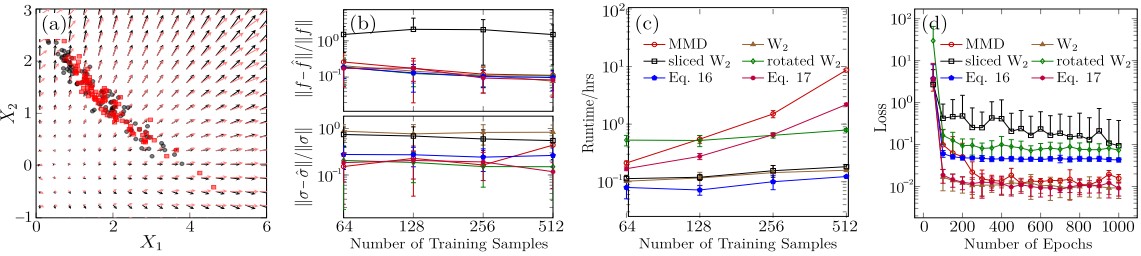

Figure 4: (a) Black and red dots are the ground-truth $(X_1(2), X_2(2))$ and the reconstructed $(\hat{X}_1(2), \hat{X}_2(2))$ found using the rotated squared $W_2$ loss function, respectively. Black and red arrows indicate, respectively, the vectors $\boldsymbol{f}(X_1, X_2)$ and $\hat{\boldsymbol{f}}(X_1, X_2)$. (b) Relative errors of the reconstructed $\boldsymbol{f}$ and $\boldsymbol{\sigma}$. Error bars indicate the standard deviation across ten reconstructions. (c) Runtime of different loss functions with respect to $N_{\text{samples}}$. (d) The decrease of different loss functions with respect to training epochs.

From Fig. 4(b), numerically evaluating Eq. 17 and the rotated-$W_2$-distance loss (the green and purple curves) perform the best, being better than the MMD method, using the sliced $W_2$ distance, or using the 2D $W_2$ loss. Numerically estimating Eq. 16 yields poorer performance than numerically estimating Eq. 17 because numerically evaluating the $W_2$ distance for high-dimensional empirical distributions is difficult and less accurate. Over training epochs, minimizing Eq. 17 leads to the fastest convergence, implying that training with Eq. 17 as the loss can be the more efficient than with other losses used in this example. On the other hand, the runtime and memory usage of numerically evaluating the time-decoupled Eq. 17 using `ot.emd2` is larger than that of the rotated $W_2$ loss when the number of training samples $N_{\text{samples}}$ is large (Fig. 4(c)), since the `ot.emd2` function needs to calculate a $N_{\text{samples}} \times N_{\text{samples}}$ cost matrix. However, using `ot.emd2` to evaluate Eq. 17 could be more advantageous than the rotated $W_2$ distance in reconstructing high-dimensional SDEs since the operations needed to calculate the cost matrix increases just linearly with dimensionality. In Appendix I, we carry out additional numerical experiments to investigate how many rotations are needed to reconstruct Eq. 22. Further analysis on the number of samples and SDE dimensionality [Fournier & Guillin (2015)] that allows $W_2$-based distances to efficiently reconstruct multivariate SDEs would be helpful.

## 4  SUMMARY & CONCLUSIONS

In this paper, we analyzed the squared $W_2$ distance and proposed a simple loss function based on it for the purpose of reconstructing SDEs. Upon performing numerical experiments, we found that our proposed squared $W_2$ distance could be more efficient than other recent machine-learning and statistical methods for SDE reconstruction. A promising future direction is to further investigate applying the squared $W_2$ loss to reconstructing high-dimensional SDEs.

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

## A  PROOF FOR THEOREM 1

First, we recall the definition of $\tilde{X}(t)$, which satisfies the following SDE

$$\mathrm{d}\tilde{X}(t) = \hat{f}(\tilde{X}(t), t)\mathrm{d}t + \hat{\sigma}(\tilde{X}(t), t)\mathrm{d}B(t), \quad \tilde{X}(0) = X(0), \tag{24}$$

where $\hat{f}, \hat{\sigma}$ are defined in Eq. 2. In other words, we choose a specific realization of $\hat{X}(t)$, *coupled to $X(t)$* in the sense that its initial values equal to $X(0)$ almost surely, and the Itô integral is defined with respect to the same standard Brownian motion $B(t)$. Therefore, by definition, if we let $\pi$ in Eq. 5 to be the joint distribution of $(X, \tilde{X})$, then

$$W_2(\mu, \hat{\mu}) \leq \left( \mathbb{E}\Big[ \int_0^T |\tilde{X}(t) - X(t)|^2 \mathrm{d}t \Big] \right)^{\frac{1}{2}}. \tag{25}$$

Next, we provide a bound for $\mathbb{E}\big[ \int_0^T |\tilde{X}(t) - X(t)|^2 \mathrm{d}t \big]^{\frac{1}{2}}$ by the mean value theorem for $f$ and $g$.

$$\mathrm{d}(X(t) - \tilde{X}(t)) = \partial_x f\big(\eta_1(X(t), \tilde{X}(t), t), t\big) \cdot (X(t) - \tilde{X}(t))\mathrm{d}t$$
$$+ \partial_x \sigma\big(\eta_2(X(t), \tilde{X}(t)), t\big) \cdot (X(t) - \tilde{X}(t))\mathrm{d}B(t) + (f - \hat{f})(\tilde{X}(t), t)\mathrm{d}t + (\sigma - \hat{\sigma})(\tilde{X}(t), t)\mathrm{d}B(t). \tag{26}$$

where $\eta_1(x_1, x_2), \eta_2(x_1, x_2)$ are defined in Theorem 1 such that their values are in $(x_1, x_2)$ and $f(X_1, t) - f(X_2, t) = \partial_x f(\eta_1(X_1, X_2), t)(X_1 - X_2)$ and $\sigma(X_1, t) - \sigma(X_2, t) = \partial_x \sigma(\eta_2(X_1, X_2), t)(X_1 - X_2)$.

We introduce an integrating factor

$$H(s, t) \coloneqq \exp\left[ \int_s^t h(X(r), \tilde{X}(r), r)\mathrm{d}r + \int_s^t \partial_x \sigma\big(\eta_2(X(r), \tilde{X}(r), r)\mathrm{d}B(r) \right], \tag{27}$$

with $h(X(t), \tilde{X}(t), t)$ defined in Eq. 9. Apply Itô's formula to $[X(t) - \tilde{X}(t)]/H(0; t)$, and we find

$$\mathrm{d}\frac{X(t) - \tilde{X}(t)}{H(0; t)} = \frac{1}{H(0; t)}\Big[ (f - \hat{f})(\tilde{X}(t), t)\mathrm{d}t + \partial_x \sigma(\eta_2(X, \tilde{X}), t) \cdot (\sigma - \hat{\sigma})(\tilde{X}(t), t)\mathrm{d}t \Big]$$
$$+ \frac{1}{H(0; t)}\Big[ (\sigma - \hat{\sigma})(\tilde{X}(t), t)\mathrm{d}B(t) \Big]. \tag{28}$$

Integrate from $0$ to $t$ on both sides to obtain

$$X(t) - \tilde{X}(t) = \int_0^t H(s;t) \cdot \Big[(f - \hat{f})(\tilde{X}(s), s) + \partial_x \sigma(\eta_2(X, \tilde{X}), s) \cdot (\sigma - \hat{\sigma})(\tilde{X}(s), s)\Big] \mathrm{d}s$$
$$+ \int_0^t H(s;t) \cdot (\sigma - \hat{\sigma})(\tilde{X}(s), s) \mathrm{d}B(s). \tag{29}$$

By invoking Itô isometry and observe that $(a + b + c)^2 \le 3(a^2 + b^2 + c^2)$, we deduce

$$\mathbb{E}[(X(t) - \tilde{X}(t))^2] \le 3\mathbb{E}\Big[\int_0^t (H(s;t) \cdot (f - \hat{f})(\tilde{X}(s), s)\mathrm{d}s)^2\Big]$$
$$+ 3\mathbb{E}\Big[(\int_0^t H(s;t) \cdot (\partial_x \sigma(\eta_2(X, \tilde{X}), s) \cdot (\sigma - \hat{\sigma})(\tilde{X}(s), s)\mathrm{d}s)^2\Big]$$
$$+ 3\mathbb{E}\Big[(\int_0^t H(s;t) \cdot (\sigma - \hat{\sigma})(\tilde{X}(s), s)\mathrm{d}B(s))^2\Big]$$
$$\le 3\mathbb{E}\Big[\int_0^t H(s;t)^2 \mathrm{d}s\Big] \times \mathbb{E}[\int_0^T (f - \hat{f})^2(\tilde{X}(s), s)\mathrm{d}s\Big]$$
$$+ 3\mathbb{E}\Big[\int_0^t H(s;t)^2 \mathrm{d}s\Big] \times \mathbb{E}[\int_0^T (\partial_x \sigma(\eta_2(X, \tilde{X}), s) \cdot (\sigma - \hat{\sigma})(\tilde{X}(s), s))^2 \mathrm{d}s\Big] \tag{30}$$
$$+ 3\mathbb{E}\Big[(\int_0^t H(s;t)^2 \times (\sigma - \tilde{\sigma})^2(\tilde{X}(s), s)\mathrm{d}s\Big]$$
$$\le 3\mathbb{E}\Big[(\int_0^t H(s;t)^2 \mathrm{d}s\Big] \times \mathbb{E}[\int_0^t (f - \hat{f})^2(\tilde{X}(s), s)\mathrm{d}s)\Big]$$
$$+ 3\mathbb{E}\Big[(\int_0^t H(s;t)^2 \mathrm{d}s\Big] \times \mathbb{E}\Big[\int_0^t (\partial_x \sigma(\eta_2(X, \tilde{X}), s) \cdot (\sigma - \hat{\sigma})(\tilde{X}(s), s))^2 \mathrm{d}s)\Big]$$
$$+ 3\mathbb{E}\Big[\int_0^t H(s;t)^4 \mathrm{d}s\Big]^{\frac{1}{2}} \times \mathbb{E}\Big[\int_0^t (\sigma - \hat{\sigma})^4(\tilde{X}(s), s)\mathrm{d}s)\Big]^{\frac{1}{2}}.$$

Finally, we conclude that

$$W_2^2(\mu, \tilde{\mu}) \le \int_0^T \mathbb{E}\big[(X(t) - \tilde{X}(t))^2\big]\mathrm{d}t$$
$$\le 3\int_0^T \mathbb{E}\Big[\int_0^t H(s;t)^2 \mathrm{d}s\Big]\mathrm{d}t \times \mathbb{E}\Big[\int_0^T (f - \hat{f})^2(\tilde{X}(s), s)\mathrm{d}s)\Big]$$
$$+ 3\int_0^T \mathbb{E}\Big[\int_0^t H(s;t)^2 \mathrm{d}s\Big]\mathrm{d}t \times \mathbb{E}\Big[\int_0^T (\partial_x \sigma(\eta_2(X, \tilde{X}), s) \cdot (\sigma - \hat{\sigma})(\tilde{X}(s), s))^2 \mathrm{d}s)\Big]$$
$$+ 3\int_0^T \Big(\mathbb{E}\Big[\int_0^t H(s;t)^4 \mathrm{d}s\Big]\Big)^{\frac{1}{2}} \mathrm{d}t \times \Big(\mathbb{E}\Big[\int_0^T (\sigma - \hat{\sigma})^4(\tilde{X}(s), s)\mathrm{d}s)\Big]\Big)^{\frac{1}{2}},$$
$$\tag{31}$$

which proves Theorem 1.

## B  Single-trajectory MSE and KL divergence

We shall first show that using the single-trajectory MSE tends to fit the mean process $\mathbb{E}[X(t)]$ and make noise diminish, which indicates that the MSE is not a good loss function when one wishes to fit $\sigma$ in Eq. 1.

For two *independent* $d$-dimensional stochastic processes $\{\boldsymbol{X}(t)\}_{t=0}^{T}, \{\hat{\boldsymbol{X}}(t)\}_{t=0}^{T}$ as solutions to Eq. 11 with appropriate $\boldsymbol{f}, \hat{\boldsymbol{f}}$ and $\boldsymbol{\sigma}, \hat{\boldsymbol{\sigma}}$, let $\mathbb{E}[\boldsymbol{X}]$ represent the trajectory of mean values of $\boldsymbol{X}(t)$, *i.e.*, $\mathbb{E}[\boldsymbol{X}] = \mathbb{E}[\boldsymbol{X}(t)]$. We have

$$
\begin{aligned}
\mathbb{E}\big[\|\boldsymbol{X} - \hat{\boldsymbol{X}}\|^2\big] = \mathbb{E}\big[\,\|\boldsymbol{X} - \mathbb{E}[\boldsymbol{X}]\|^2\,\big] + \mathbb{E}\big\|\hat{\boldsymbol{X}} - \mathbb{E}[\boldsymbol{X}]\|^2\big] \\
- 2\mathbb{E}\left[\int_0^T \Big(\boldsymbol{X} - \mathbb{E}[\boldsymbol{X}], \hat{\boldsymbol{X}} - \mathbb{E}[\boldsymbol{X}]\Big)\,\mathrm{d}t\right],
\end{aligned} \tag{32}
$$

where $\|\boldsymbol{X}\|^2 := \int_0^T |\boldsymbol{X}|_2^2 \mathrm{d}t$, $|\cdot|_2$ denotes the $\ell^2$ norm of a vector, and $(\cdot, \cdot)$ is the inner product of two $d$-dimensional vectors. In view of the independence between $\boldsymbol{X} - \mathbb{E}[\boldsymbol{X}]$ and $\hat{\boldsymbol{X}} - \mathbb{E}[\boldsymbol{X}]$, we have $\mathbb{E}\left[\Big(\boldsymbol{X} - \mathbb{E}[\boldsymbol{X}], \hat{\boldsymbol{X}} - \mathbb{E}[\boldsymbol{X}]\Big)\right] = \mathbb{E}\big[\,(\boldsymbol{X} - \mathbb{E}[\boldsymbol{X}])\,\big] \cdot \mathbb{E}\left[\Big(\hat{\boldsymbol{X}} - \mathbb{E}[\boldsymbol{X}]\Big)\right] = 0$, and

$$
\mathbb{E}\|\boldsymbol{X} - \hat{\boldsymbol{X}}\|^2 \geq \mathbb{E}\,\|\boldsymbol{X} - \mathbb{E}[\boldsymbol{X}]\|^2. \tag{33}
$$

Therefore, the optimal $\hat{\boldsymbol{X}}$ that minimizes the MSE is $\hat{\boldsymbol{X}} = \mathbb{E}[\boldsymbol{X}]$, which indicates that the MSE tends to fit the mean process $\mathbb{E}[\boldsymbol{X}]$ and make noise diminish. This is not desirable when one wishes to fit a nonzero $\sigma$ in Eq. 1.

The KL divergence, in some cases, will diverge and thus is not suitable for being used as a loss function. Here, we provide a simple intuitive example when the KL divergence fail. If we consider the degenerate case when $\mathrm{d}X(t) = \mathrm{d}t, \mathrm{d}\hat{X}(t) = (1 - \epsilon)\mathrm{d}t, t \in [0, T]$, then $D_{KL}(\mu, \hat{\mu}) = \infty$ no matter how small $\epsilon \neq 0$ is because $\mu, \hat{\mu}$ has different and degenerate support. However, from Theorem 1, $\lim_{\epsilon \to 0} W_2(\mu, \hat{\mu}) = 0$. Therefore, the KL divergence cannot effectively measure the similarity between $\mu, \hat{\mu}$. Overall, the squared $W_2$ distance is a better metric than some of the commonly used loss metrics such as the MSE or the KL divergence.

## C  Proof for Theorem 2

Here we provide proof for Theorem 2. We denote

$$
\Omega_N := \{\boldsymbol{Y}(t)|\boldsymbol{Y}(t) = \boldsymbol{Y}(t_i) \ t \in [t_i, t_{i+1}), i < N - 1; \ \boldsymbol{Y}(t) = \boldsymbol{Y}(t_i), \ t \in [t_i, t_{i+1}]\} \tag{34}
$$

to be the space of piecewise functions. We also define the space

$$
\tilde{\Omega}_N := \{\boldsymbol{Y}_1(t) + \boldsymbol{Y}_2(t), \boldsymbol{Y}_1 \in C([0, T]; \mathbb{R}^d), \boldsymbol{Y}_2 \in \Omega_N\}. \tag{35}
$$

$\tilde{\Omega}_N$ is also a seperable metric space because both $\big(C([0, T]; \mathbb{R}^d), \|\cdot\|\big)$ and $\big(\Omega_N, \|\cdot\|\big)$ are separable metric spaces. Furthermore, both the embedding mapping from $C([0, T]; \mathbb{R}^d)$ to $\tilde{\Omega}_N$ and the embedding mapping from $\Omega_N$ to $\tilde{\Omega}_N$ preserves the $\|\cdot\|$ norm. They, the two embedding mappings are measurable, which enables us to define the measures on $\mathcal{B}(\tilde{\Omega}_N)$ induced by the measures $\mu, \hat{\mu}$ on $\mathcal{B}\big(C([0, T]; \mathbb{R}^d)\big)$ and the measures $\mu_N, \hat{\mu}_N$ on $\mathcal{B}(\Omega_N)$. For notational simplicity, we shall still denote those induced measures by $\mu, \hat{\mu}, \mu_N, \hat{\mu}_N$.

Therefore, the inequality Eq. 13 is a direct result of the triangular inequality for the Wasserstein distance (Clement & Desch, 2008) because $\boldsymbol{X}, \boldsymbol{X}_N, \hat{\boldsymbol{X}}, \hat{\boldsymbol{X}}_N \in \tilde{\Omega}_N$.

Next, we shall prove Eq. 15 when $\boldsymbol{X}(t)$, $\hat{\boldsymbol{X}}(t)$ are solutions to SDEs Eq. 1 and Eq. 2. Because $\boldsymbol{X}_N(t)$ is the projection to $\boldsymbol{X}(t)$, the squared $W_2^2(\mu, \mu_N)$ can be bounded by

$$W_2^2(\mu, \mu_N) \leq \sum_{i=1}^{N} \int_{t_{i-1}}^{t_i} \mathbb{E}\big[|\boldsymbol{X}(t) - \boldsymbol{X}_N(t)|_2^2\big]\mathrm{d}t = \sum_{i=1}^{N} \int_{t_{i-1}}^{t_i} \sum_{\ell=1}^{d} \mathbb{E}\big[(X_\ell(t) - X_{N,\ell}(t))^2\big]\mathrm{d}t \tag{36}$$

For the first inequality above, we choose a specific *coupling*, i.e. the coupled distribution, $\pi$ of $\mu, \mu_N$ that is essentially the "original" probability distribution. To be more specific, for an abstract probability space $(\Omega, \mathcal{A}, p)$ associated with $\boldsymbol{X}$, $\mu$ and $\mu_N$ can be characterized by the *pushforward* of $p$ via $\boldsymbol{X}$ and $\boldsymbol{X}_N$ respectively, i.e., $\mu = \boldsymbol{X}_* p$, defined by $\forall A \in \mathcal{B}(\tilde{\Omega}_N)$, elements in the Borel $\sigma$-algebra of $\tilde{\Omega}_N$,

$$\mu(A) = \boldsymbol{X}_* p(A) \coloneqq p(\boldsymbol{X}^{-1}(A)), \tag{37}$$

where $\boldsymbol{X}$ is interpreted as a measurable map from $\Omega$ to $\tilde{\Omega}_N$, and $\boldsymbol{X}^{-1}(A)$ is the preimage of $A$ under $\boldsymbol{X}$. Then, the coupling $\pi$ is defined by

$$\pi = (\boldsymbol{X}, \boldsymbol{X}_N)_* p, \tag{38}$$

where $(\boldsymbol{X}, \boldsymbol{X}_N)$ is interpreted as a measurable map from $\Omega$ to $\tilde{\Omega}_N \times \tilde{\Omega}_N$. One can readily verify that the marginal distributions of $\pi$ are $\mu$ and $\mu_N$ respectively. Recall that $s$ represents the dimension of the standard Brownian motions in the SDEs.

For each $\ell = 1, ..., d$, we have

$$\sum_{i=1}^{N} \int_{t_{i-1}}^{t_i} \mathbb{E}\big[(X_\ell(t) - X_{N,\ell}(t))^2\big]\mathrm{d}t$$

$$\leq (s+1)\left[\sum_{i=1}^{N} \int_{t_{i-1}}^{t_i} \left(\mathbb{E}\Big[\Big(\int_{t_i}^{t} f_\ell(\hat{X}(r), r)\mathrm{d}r\Big)^2\Big] + \mathbb{E}\Big[\Big(\int_{t_i}^{t} \sum_{j=1}^{s} \sigma_{\ell,j}(\hat{X}(r), r)\mathrm{d}B_j(r)\Big)^2\Big]\right)\mathrm{d}t\right]$$

$$\leq (s+1)\sum_{i=1}^{N} \left((\Delta t)^2 \mathbb{E}\Big[\int_{t_{i-1}}^{t_i} f_\ell^2 \mathrm{d}t\Big] + \Delta t \sum_{j} \mathbb{E}\Big[\int_{t_{i-1}}^{t_i} \sigma_{\ell,j}^2 \mathrm{d}t\Big]\right) \tag{39}$$

The first inequality follows from the observation that $(\sum_{i=1}^{n} a_i)^2 \leq n(\sum_{i=1}^{n} a_i^2)$ and application of this observation to the integral representation of $\boldsymbol{X}(t)$.

Summing over $\ell$, we have

$$\Big(\sum_{i=1}^{N} \int_{t_{i-1}}^{t_i} \mathbb{E}\big[|\boldsymbol{X}(t) - \boldsymbol{X}_N(t)|_2^2\big]\mathrm{d}t\Big)^{\frac{1}{2}} \leq \sqrt{s+1}\big(F(\Delta t)^2 + \Sigma \Delta t\big)^{\frac{1}{2}} \tag{40}$$

Similarly, $W_2(\hat{\mu}, \hat{\mu}_N)$ can be bounded by

$$W_2(\hat{\mu}, \hat{\mu}_N) \leq \sqrt{s+1}\big(\hat{F}(\Delta t)^2 + \hat{\Sigma}\Delta t\big)^{\frac{1}{2}}. \tag{41}$$

Plug Eq. 40 and Eq. 41 into Eq. 13, we have proved Eq. 15.

## D    DEFINITION OF DIFFERENT LOSS METRICS USED IN THE EXAMPLES

Five loss functions for 1D cases were considered as follows:

1. Squared 2-Wasserstein distance averaged over each time step:

$$\tilde{W}_2^2(\mu_N, \hat{\mu_N}) = \sum_{i=1}^{N-1} W_2^2(\mu_{N,i}, \hat{\mu}_{N,i})\Delta t,$$

   where $\Delta t$ is the time step size, and $W_2$ is the 2-Wasserstein distance between two empirical distributions $\mu_{N,i}, \hat{\mu}_{N,i}$, where the two empirical distributions $\mu_{N,i}, \hat{\mu}_{N,i}$ are calculated by the samples of the trajectories at a given time step $t_i$.

2. Mean squared error (MSE) between the trajectories, where $M$ is the total number of the ground-truth and prediction trajectories. $x_{i,j}$ and $\hat{x}_{i,j}$ are the values of the $j^{\text{th}}$ ground-truth and prediction trajectories at time $t_i$, respectively:

$$\text{MSE}(X, \widehat{X}) = \sum_{i=1}^{N} \sum_{j=1}^{M} (x_{i,j} - \hat{x}_{i,j})^2 \Delta t.$$

3. The sum of squared distance between mean trajectories and absolute distance between trajectories, which is a common practice for estimating the parameters of an SDE. Here $M$ and $x_{i,j}$ and $\hat{x}_{i,j}$ have the same meaning as in the MSE definition. $\text{var}(X_i)$ and $\text{var}(\hat{X}_i)$ are the variances of the empirical distributions of $X(t_i), \hat{X}(t_i)$, respectively. We shall denote this loss function by

$$(\text{Mean}^2 + \text{Var})(X, \hat{X}) = \sum_{i=1}^{N} \left[ \left( \frac{1}{n} \sum_{j=1}^{M} x_{i,j} - \frac{1}{n} \sum_{i=1}^{N} \hat{x}_{i,j} \right)^2 + \left| \text{var}(X_i) - \text{var}(\hat{X}_i) \right| \right] \Delta t.$$

4. Negative approximate log-likelihood of the trajectories:

$$-\log \mathcal{L}(X|\sigma) = -\sum_{i=0}^{N-1} \sum_{j=1}^{M} \log \rho_{\mathcal{N}} \left[ \frac{x_{i+1,j} - x_{t,j} + f(x_{i,j}, t_i)\Delta t}{\sigma^2(x_{i,j}, t_i)\Delta t} \right],$$

   where $\rho_{\mathcal{N}}$ stands for the probability density function of the standard normal distribution, and $f(x_{i,j}, t_i), \sigma(x_{i,j}, t_i)$ are the ground-truth drift and diffusion functions in Eq. 1. $M$ and $x_{i,j}$ and $\hat{x}_{i,j}$ have the same meaning as in the MSE definition.

5. MMD (maximum mean discrepancy) (Li et al., 2015):

$$\text{MMD}(X, \hat{X}) = \sum_{i=1}^{N} \left( \mathbb{E}_p[K(X_i, X_i)] - 2\mathbb{E}_{p,q}[K(X_i, \hat{X}_i)] + \mathbb{E}_q[K(\hat{X}_i, \hat{X}_i)] \right) \Delta t,$$

   where $K$ is the standard radial basis function (or Gaussian kernel) with multiplier 2 and number of kernels 5.

Five $W_2$ distance based loss functions for the 2D SDE reconstruction problem Example 3.4 are listed as follows

1. 2D squared $W_2$ loss

$$\sum_{i=1}^{N-1} \left( W_2^2(\mu_{N,i,1}, \hat{\mu}_{N,i,1}) + W_2^2(\mu_{N,i,2}, \hat{\mu}_{N,i,2}) \right) \Delta t \tag{42}$$

   where $\mu_{N,i,1}$ and $\hat{\mu}_{N,i,1}$ are the empirical distributions of $X_1, \hat{X}_1$ at time $t_i$, respectively. Also, $\mu_{N,i,2}$ and $\hat{\mu}_{N,i,2}$ are the empirical distributions of $X_2, \hat{X}_2$ at time $t_i$, respectively.

2. Weighted sliced squared $W_2$ loss

$$\sum_{i=1}^{N-1} \Big( \sum_{k=1}^{m} \frac{N_k}{\sum_{k=1}^{m} N_k} W_2^2(\mu_{N,i,k}^{\mathrm{s}}, \hat{\mu}_{N,i,k}^{\mathrm{s}}) \Big) \Delta t$$

where $\mu_{N,i,k}^{\mathrm{s}}$ is the empirical distribution for $\sqrt{X_1(t_i)^2 + X_2(t_i)^2}$ such that the angle between the two vectors $(X_1(t_i), X_2(t_i))$ and $(1,0)$ is in $[\frac{2(k-1)\pi}{m}, \frac{2k\pi}{m})$; $\hat{\mu}_{N,i,k}^{\mathrm{s}}$ is the empirical distribution for $\sqrt{\hat{X}_1(t_i)^2 + \hat{X}_2(t_i)^2}$ such that the angle between the two vectors $(\hat{X}_1(t_i), \hat{X}_2(t_i))$ and $(1,0)$ is in $[\frac{2(k-1)\pi}{m}, \frac{2k\pi}{m})$; $N_k$ is the number of predictions such that the angle between the two vectors $(\hat{X}_1(t_i), \hat{X}_2(t_i))$ and $(1,0)$ is in $[\frac{2(k-1)\pi}{m}, \frac{2k\pi}{m})$.

3. The loss function Eq. 16

$$W_2^2(\mu_N^{\mathrm{e}}, \hat{\mu}_N^{\mathrm{e}}),$$

where $\mu_N^{\mathrm{e}}$ and $\hat{\mu}_N^{\mathrm{e}}$ are the empirical distributions of the vector $(\boldsymbol{X}(t_1), ... \boldsymbol{X}(t_{N-1}))$ and $(\hat{\boldsymbol{X}}(t_1), ..., \hat{\boldsymbol{X}}(t_{N-1}))$, respectively. It is estimated by

$$W_2^2(\mu_N^{\mathrm{e}}, \hat{\mu}_N^{\mathrm{e}}) \approx \mathtt{ot.emd2}(\frac{1}{M}\boldsymbol{I}_M, \frac{1}{M}\boldsymbol{I}_M, \boldsymbol{C}), \tag{43}$$

where $\mathtt{ot.emd2}$ is the function for solving the earth movers distance problem in the $\mathtt{ot}$ package of Python, $M$ is the number of ground-truth and predicted trajectories, $\boldsymbol{I}_\ell$ is an $M$-dimensional vector whose elements are all 1, and $\boldsymbol{C} \in \mathbb{R}^{M \times M}$ is a matrix with entries $(\boldsymbol{C})_{ij} = |\boldsymbol{X}_N^i - \hat{\boldsymbol{X}}_N^j|_2^2$. $\boldsymbol{X}_N^i$ is the vector of the values of the $i^{\mathrm{th}}$ ground-truth trajectory at time points $t_1, ..., t_{N-1}$, and $\hat{\boldsymbol{X}}_N^j$ is the vector of the values of the $j^{\mathrm{th}}$ predicted trajectory at time points $t_1, ..., t_{N-1}$.

4. The formula on the right-hand-side of Eq. 17, which is an approximation of Eq. 16. It is estimated by

$$\sum_{i=1}^{N-1} \inf_{\pi_i} \mathbb{E}_{\pi_i}[|\boldsymbol{X}(t_i) - \hat{\boldsymbol{X}}(t_i)|_2^2] \Delta t$$

$$\approx \sum_{i=1}^{N-1} W_2^2(\mu_{N,i}^{\mathrm{e}}, \hat{\mu}_{N,i}^{\mathrm{e}}) \Delta t \approx \Delta t \sum_{i=1}^{N-1} \mathtt{ot.emd2}(\frac{1}{M}\boldsymbol{I}_M, \frac{1}{M}\boldsymbol{I}_M, \boldsymbol{C}_i), \tag{44}$$

where $\mu_{N,i}^{\mathrm{e}}, \hat{\mu}_{N,i}^{\mathrm{e}}$ are the empirical distribution of $\boldsymbol{X}(t_i)$, $\hat{\boldsymbol{X}}(t_i)$, respectively, and $\mathtt{ot.emd2}$ is the function for solving the earth movers distance problem in the $\mathtt{ot}$ package of Python, $M$ is the number of ground-truth and predicted trajectories, $\boldsymbol{I}_M$ is an $\ell$-dimensional vector whose elements are all 1. Here, the matrix $\boldsymbol{C}_i \in \mathbb{R}^{M \times M}$ has entries $(\boldsymbol{C}_i)_{sj} = |\boldsymbol{X}^s(t_i) - \hat{\boldsymbol{X}}^j(t_i)|_2^2$ for $i = 1, ..., N-1$. $\boldsymbol{X}^s(t_i)$ is the vector of the values of the $s^{\mathrm{th}}$ ground-truth trajectory at the time point $t_i$, and $\hat{\boldsymbol{X}}^j(t_i)$ is the vector of the values of the $j^{\mathrm{th}}$ predicted trajectory at the time point $t_i$.

5. Rotated squared $W_2$ loss

$$\sum_{k=0}^{m-1} \sum_{i=1}^{N-1} \Big( W_2^2(\mu_{N,i,1}^k, \hat{\mu}_{N,i,1}^k) + W_2^2(\mu_{N,i,2}^k, \hat{\mu}_{N,i,2}^k) \Big) \Delta t \tag{45}$$

where $\mu_{N,i,1}^k$ is the empirical distribution for the random variable $X_1^k(t_i) := \big( \cos(\frac{k\pi}{2m}) X_1(t_i) + \sin(\frac{(k-1)\pi}{2m}) X_2(t_i) \big)$ and $\mu_{N,i,2}^k$ is the empirical distribution for the random variable $X_2^k := \big( -\sin(\frac{k\pi}{2m}) X_1(t_i) + \cos(\frac{k\pi}{2m}) X_2(t_i) \big)$. Similarly, $\hat{\mu}_{N,i,1}^k$ is the empirical distribution for the random variables $\hat{X}_1^k := \big( \cos(\frac{k\pi}{2m}) \hat{X}_1(t_i) + \sin(\frac{k\pi}{2m}) \hat{X}_2(t_i) \big)$ and $\hat{\mu}_{N,i,2}^k$ is the empirical distribution

for the random variable $\hat{X}_2^k := \big( -\sin(\frac{(k-1)\pi}{2m})\hat{X}_1(t_i) + \cos(\frac{k\pi}{2m})\hat{X}_2(t_i) \big)$. This rotated squared $W_2$ loss generalizes the 2D squared $W_2$ loss by rotating the vector $(X_1, X_2)$ and $(\hat{X}_1, \hat{X}_2)$ to the same degree $\theta$, $\theta = \frac{k\pi}{2m}$, $k = 0, ..., m - 1$ and projecting the rotated vectors into $X_1$ and $X_2$ axes, aiming at providing a potential way to resolve the correlation between $X_1, X_2$. The 2D squared $W_2$ loss Eq. 42 may not effectively handle skewed (correlated) distributions, which can be addressed by applying a suitable rotation and using the rotated squared $W_2$ loss. A figure of such a 2D skewed distribution and how a rotation can help identify the correlation between the two dimensions is shown here. For linear rotations, the original and rotated SDEs can interconvert with each

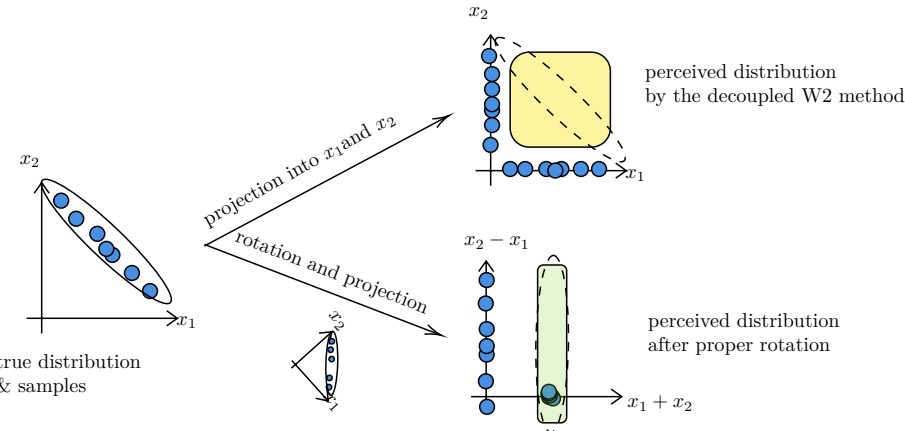

Figure S1: An illustration of how a proper rotation helps resolve the correlation between $(x_1, x_2)$ for a 2-dimensional distribution.

other reversibly, without loss of information. Specifically, the rotated squared $W_2$ loss (Eq. 45) is a lower bound for $m \sum_{i=1}^{N-1} W_2^2(\mu_{N,i}^e, \hat{\mu}_{N,i}^e) \Delta t$, where $\mu_{N,i}^e, \hat{\mu}_{N,i}^e$ are empirical distributiosn for $\boldsymbol{X}(t_i), \hat{\boldsymbol{X}}(t_i)$, respectively. Actually, in Eq. 45, for each $k$ and $i$,

$$W_2^2(\mu_{N,i,1}^k, \hat{\mu}_{N,i,1}^k) + W_2^2(\mu_{N,i,2}^k, \hat{\mu}_{N,i,2}^k) \leq W_2^2(\mu_{N,i}^e, \hat{\mu}_{N,i}^e). \tag{46}$$

To be more specific, if $\pi_i(\mu_{N,i}^e, \hat{\mu}_{N,i}^e)$ is a coupling such that its marginal distributions satisfy

$$\begin{aligned}
\int \pi_i(\mu_{N,i}^e, \hat{\mu}_{N,i}^e)\big(\boldsymbol{X}(t_i), \hat{\boldsymbol{X}}(t_i)\big)\mathrm{d}\boldsymbol{X}(t_i) &= \hat{\mu}_{N,i}^e(\hat{\boldsymbol{X}}(t_i)), \\
\int \pi_i(\mu_{N,i}^e, \hat{\mu}_{N,i}^e)\big(\boldsymbol{X}(t_i), \hat{\boldsymbol{X}}(t_i)\big)\mathrm{d}\hat{\boldsymbol{X}}(t_i) &= \mu_{N,i}^e(\boldsymbol{X}(t_i)),
\end{aligned} \tag{47}$$

then the marginal distributions of $\pi_i(\mu_{N,i}^{\mathrm{e}}, \hat{\mu}_{N,i}^{\mathrm{e}})$ also satisfy

$$\int \pi_i(\mu_{N,i}^{\mathrm{e}}, \hat{\mu}_{N,i}^{\mathrm{e}})\big(X_1^k(t_i), X_2^k(t_i), \hat{X}_1^k(t_i), \hat{X}_2^k(t_i)\big)\mathrm{d}X_1^k(t_i)\mathrm{d}\hat{X}_1^k(t_i)\mathrm{d}\hat{X}_2^k(t_i) = \mu_{N,i,2}^k(X_2^k(t_i)),$$

$$\int \pi_i(\mu_{N,i}^{\mathrm{e}}, \hat{\mu}_{N,i}^{\mathrm{e}})\big(X_1^k(t_i), X_2^k(t_i), \hat{X}_1^k(t_i), \hat{X}_2^k(t_i)\big)\mathrm{d}X_2^k(t_i)\mathrm{d}\hat{X}_1^k(t_i)\mathrm{d}\hat{X}_2^k(t_i) = \mu_{N,i,1}^k(X_1^k(t_i))$$

$$\int \pi_i(\mu_{N,i}^{\mathrm{e}}, \hat{\mu}_{N,i}^{\mathrm{e}})\big(X_1^k(t_i), X_2^k(t_i), \hat{X}_1^k(t_i), \hat{X}_2^k(t_i)\big)\mathrm{d}X_1^k(t_i)\mathrm{d}X_2^k(t_i)\mathrm{d}\hat{X}_1^k(t_i) = \hat{\mu}_{N,i,1}^k(\hat{X}_2^k(t_i)),$$

$$\int \pi_i(\mu_{N,i}^{\mathrm{e}}, \hat{\mu}_{N,i}^{\mathrm{e}})\big(X_1^k(t_i), X_2^k(t_i), \hat{X}_1^k(t_i), \hat{X}_2^k(t_i)\big)\mathrm{d}X_1^k(t_i)\mathrm{d}X_2^k(t_i)\mathrm{d}\hat{X}_2^k(t_i) = \hat{\mu}_{N,i,2}^k(\hat{X}_1^k(t_i)).$$
$$\tag{48}$$

Thus, for all $\pi_i$ that satisfies the condition Eq. 47, we conclude that

$$W_2^2(\mu_{N,i,1}^k, \hat{\mu}_{N,i,1}^k) + W_2^2(\mu_{N,i,2}^k, \hat{\mu}_{N,i,2}^k) \le \mathbb{E}_{\pi_i}\big[|X_1^k(t_i) - \hat{X}_1^k(t_i)|^2 + |X_2^k(t_i) - \hat{X}_2^k(t_i)|^2\big]$$
$$= \mathbb{E}_{\pi_i}\big[|\boldsymbol{X}(t_i) - \hat{\boldsymbol{X}}(t_i)|_2^2\big],$$
$$\tag{49}$$

which implies

$$W_2^2(\mu_{N,i,1}^k, \hat{\mu}_{N,i,1}^k) + W_2^2(\mu_{N,i,2}^k, \hat{\mu}_{N,i,2}^k) \le \inf_{\pi_i} E_{\pi_i}\big[|\boldsymbol{X}(t_i) - \hat{\boldsymbol{X}}(t_i)|_2^2\big] = W_2^2(\mu_{N,i}^{\mathrm{e}} \hat{\mu}_{N,i}^{\mathrm{e}}). \tag{50}$$

## E  DEFAULT TRAINING SETTING

Here we list the default training hyperparameters and gradient descent methods for each example in Table 1.

| Loss | Example 1 | Example 2 | Example 3 | Example 4 | Appendix K |
|---|---|---|---|---|---|
| Gradient descent method | AdamW | AdamW | AdamW | AdamW | AdamW |
| Learning rate | 0.001 | 0.002 | 0.002 | 0.0005 | 0.002 |
| Weight decay | 0.005 | 0.005 | 0.005 | 0.005 | 0.005 |
| Number of epochs | 1000 | 2000 | 2000 | 2000 | 500 |
| Number of samples | 100 | 200 | 256 | 200 | 100 |
| Hidden layers in $\Theta_1$ | 2 | 1 | 1 | 1 | 1 |
| Neurons in each layer in $\Theta_1$ | 32 | 32 | 32 | 32 | 150 |
| Hidden layers in $\Theta_2$ | 2 | 1 | 1 | 1 | 1 |
| Activation function | tanh | ReLu | ReLu | ReLu | ReLu |
| Neurons in each layer in $\Theta_2$ | 32 | 32 | 32 | 32 | 150 |
| $\Delta t$ | 0.1 | 0.05 | 1 | 0.02 | 0.5 |

Table 1: Training settings for each example.

## F  UNCERTAINTY IN THE INITIAL CONDITION

For reconstructing the CIR model Eq. 20 in Example 3.2, instead of using the same initial condition for all trajectories, we shall sample the initial condition from a distribution to investigate the numerical performance of our proposed squared $W_2$ distance loss when the initial condition is not fixed and instead is sampled from a distribution.

First, we construct an additional dataset of the CIR model to allow the initial value $u_0 \sim \mathcal{N}(2, \delta^2)$, with $\delta^2$ ranging from 0 to 1, and $\mathcal{N}$ stands for the 1D normal distribution. We then train the model by minimizing Eq. 17 to reconstruct Eq. 20 with the same hyperparameters as in Example 3.2. The results are shown in Table 2, which indicate our proposed squared $W_2$ loss function is rather insensitive to the "noise", *i.e.*, the variance in the distribution of the initial condition.

| Loss | $\delta$ | Relative Errors in $f$ | Relative Errors in $\sigma$ | $N_{\text{repeats}}$ |
|------|------|------|------|------|
| $W_2$ | 0.0 | 0.072 ($\pm$ 0.008) | 0.071 ($\pm$ 0.023) | 10 |
| $W_2$ | 0.1 | 0.053 ($\pm$ 0.008) | 0.043 ($\pm$ 0.016) | 10 |
| $W_2$ | 0.2 | 0.099 ($\pm$ 0.007) | 0.056 ($\pm$ 0.019) | 10 |
| $W_2$ | 0.3 | 0.070 ($\pm$ 0.014) | 0.083 ($\pm$ 0.026) | 10 |
| $W_2$ | 0.4 | 0.070 ($\pm$ 0.014) | 0.078 ($\pm$ 0.040) | 10 |
| $W_2$ | 0.5 | 0.075 ($\pm$ 0.013) | 0.138 ($\pm$ 0.021) | 10 |
| $W_2$ | 0.6 | 0.037 ($\pm$ 0.018) | 0.069 ($\pm$ 0.017) | 10 |
| $W_2$ | 0.7 | 0.075 ($\pm$ 0.016) | 0.043 ($\pm$ 0.014) | 10 |
| $W_2$ | 0.8 | 0.041 ($\pm$ 0.012) | 0.079 ($\pm$ 0.023) | 10 |
| $W_2$ | 0.9 | 0.082 ($\pm$ 0.015) | 0.108 ($\pm$ 0.033) | 10 |
| $W_2$ | 1.0 | 0.058 ($\pm$ 0.024) | 0.049 ($\pm$ 0.025) | 10 |

Table 2: Reconstructing the CIR model Eq. 20 when $u_0 \sim \mathcal{N}(2, \delta^2)$ with different variance $\delta^2$. The results indicate that the reconstruction results are not sensitive to the variance in the distribution of the initial value $u_0$.

## G  NEURAL NETWORK STRUCTURE

We examine how the neural network structure affects the reconstruction of the CIR model Eq. 20 in Example 3.2. We vary the number of layers and the numbqer of neurons in each layer (the number of neurons are set to be the same in each hidden layer), and the results are shown in Table 3.

The results in Table 3 show that increasing the number of neurons in each layer improves the reconstruction accuracy in $\sigma$. For the reconstructing CIR model in Example 3.2, using 32 neurons in each layer seems to be sufficient. On the other hand, when each layer contains 32 neurons, the number of hidden layers in the neural

Table 3: Reconstructing the CIR model when using neuron networks of different widths and numbers in each hidden layer to parameterize $\hat{f}, \hat{\sigma}$ in Eq. 2.

| Loss | Width | Layer | Relative Errors in $f$ | Relative Errors in $\sigma$ | $N_{\text{repeats}}$ |
|------|------|------|------|------|------|
| $W_2$ | 16 | 1 | 0.131($\pm$0.135) | 0.170($\pm$0.102) | 10 |
| $W_2$ | 32 | 1 | 0.041($\pm$0.008) | 0.109($\pm$0.026) | 10 |
| $W_2$ | 64 | 1 | 0.040($\pm$0.008) | 0.104($\pm$0.019) | 10 |
| $W_2$ | 128 | 1 | 0.040($\pm$0.008) | 0.118($\pm$0.019) | 10 |
| $W_2$ | 32 | 2 | 0.049($\pm$0.015) | 0.123($\pm$0.020) | 10 |
| $W_2$ | 32 | 3 | 0.094($\pm$0.013) | 0.166($\pm$0.041) | 10 |
| $W_2$ | 32 | 4 | 0.124($\pm$0.020) | 0.185($\pm$0.035) | 10 |
| $W_2$ | 32 | 5 | 0.041($\pm$0.008) | 0.122($\pm$0.024) | 10 |
| $W_2$ | 32 | 6 | 0.043($\pm$0.013) | 0.117($\pm$0.024) | 10 |
| $W_2$ | 32 | 7 | 0.044($\pm$0.012) | 0.109($\pm$0.017) | 10 |

Table 4: Reconstructing the CIR model Eq. 20 when neuron networks have different numbers of hidden layers and are equipped with the ResNet technique. Each hidden layer contains 32 neurons.

| Loss | Layer | Relative Errors in $f$ | Relative Errors in $\sigma$ | $N_{\text{repeats}}$ |
|------|-------|------------------------|----------------------------|----------------------|
| $W_2$ | 1 | $0.045(\pm0.012)$ | $0.116(\pm0.025)$ | 10 |
| $W_2$ | 2 | $0.053(\pm0.011)$ | $0.108(\pm0.024$ | 10 |
| $W_2$ | 3 | $0.071(\pm0.017)$ | $0.117(\pm0.040)$ | 10 |
| $W_2$ | 4 | $0.096(\pm0.035)$ | $0.149(\pm0.064)$ | 10 |

network seems does not affect the reconstruction accuracy of $f, \sigma$, and this indicates even 1 or 2 hidden layers are sufficient for the reconstruction of $f, \sigma$. Thus, reconstructing the CIR model in Example 3.2 using our proposed squared $W_2$ based loss function does not require using complex deep or wide neural networks.

We also consider using the ResNet neural network structure (He et al., 2016). However, the application of the ResNet technique does not improve the reconstruction accuracy of the CIR model in Example 3.2. This is because simple feedforward multilayer neural network structure could work well for learning Eq. 20 when reconstructing both $f$ and $\sigma$ so we do not need deep neural networks. Thus. the Resnet technique is not required. The results are shown in Table 4.

## H  USING THE STOCHASTIC GRADIENT DESCENT METHOD FOR OPTIMIZATION

Here, we shall reconstruct the OU process Eq. 21 in Example 3.3 with the initial condition $X(0) = 0$ using the MMD and our squared $W_2$ distance as loss functions with different numbers of ground-truth trajectories and different batch sizes for applying the stochastic gradient descent technique for optimizing the parameters in the neural networks for reconstructing the SDE. We train 2000 epochs with a learning rate 0.001 for all

Table 5: Errors and runtime for different loss functions and different numbers of ground-truth trajectories when the training batch size is fixed to 16 and 256. The MMD and our proposed squared $W_2$ distance Eq. 17 are used as the loss function.

| Loss | $N_{\text{sample}}$ | Relative Error in $f$ | Relative Error in $\sigma$ | Runtime | $N_{\text{repeats}}$ | Batch Size |
|------|---------------------|-----------------------|---------------------------|---------|----------------------|------------|
| MMD | 256 | $0.25 \pm 0.09$ | $0.43 \pm 0.18$ | $0.43 \pm 0.04$ | 10 | 16 |
| MMD | 512 | $0.29 \pm 0.08$ | $0.41 \pm 0.19$ | $0.37 \pm 0.0$ | 10 | 16 |
| MMD | 1024 | $0.27 \pm 0.09$ | $0.41 \pm 0.21$ | $0.37 \pm 0.0$ | 10 | 16 |
| $W_2$ | 256 | $0.21 \pm 0.07$ | $0.41 \pm 0.13$ | $0.42 \pm 0.02$ | 10 | 16 |
| $W_2$ | 512 | $0.21 \pm 0.06$ | $0.38 \pm 0.16$ | $0.36 \pm 0.00$ | 10 | 16 |
| $W_2$ | 1024 | $0.19 \pm 0.05$ | $0.41 \pm 0.15$ | $0.36 \pm 0.00$ | 10 | 16 |
| MMD | 256 | $0.24 \pm 0.10$ | $2.71 \pm 6.15$ | $1.60 \pm 0.29$ | 10 | 256 |
| MMD | 512 | $0.21 \pm 0.15$ | $0.38 \pm 0.22$ | $1.81 \pm 0.13$ | 10 | 256 |
| MMD | 1024 | $0.21\pm 0.13$ | $0.33 \pm 0.20$ | $1.82 \pm 0.13$ | 10 | 256 |
| $W_2$ | 256 | $0.21 \pm 0.10$ | $0.40 \pm 0.20$ | $0.63 \pm 0.05$ | 10 | 256 |
| $W_2$ | 512 | $0.24 \pm 0.10$ | $0.34 \pm 0.18$ | $0.49 \pm 0.003$ | 10 | 256 |
| $W_2$ | 1024 | $0.23 \pm 0.08$ | $0.32 \pm 0.16$ | $0.49 \pm 0.005$ | 10 | 256 |

numerical experiments, which is sufficient for all cases because the loss function stays almost unchanged before 2000 epochs. From Table 5, for both the MMD and the squared $W_2$ distance, a larger number of training samples leads to more accurate reconstruction of $\sigma$ (the noise term). Furthermore, it can be seen from Table 5 that using a smaller batch size (16) for training leads to inaccurate reconstruction of $\sigma$ for

applying both the MMD and the squared $W_2$ distance as the loss function even if the number of trajectories in the training set is large. This might be owing to the fact that the trajectories are intrinsically noisy, and thus in order to reconstruct the noise in the SDE Eq. 17, a larger batch size is needed to capture the statistical properties of the noise. Therefore, using a smaller batch size could not help remedy the high cost of MMD as the reconstruction error is large and leads to inaccurate reconstruction of the ground-truth SDE. Thus, a larger batch size for training could be an intrinsic need for accurately reconstructing SDEs.

From both the results in Example 3.3 and Table 5, our proposed squared $W_2$ distance is faster and more efficient than the MMD method, making it potentially more suitable than the MMD loss function for reconstructing SDEs.

# I  SENSITIVITY TEST FOR USING THE ROTATED SQUARED $W_2$ DISTANCE LOSS FUNCTION

We experimented with different numbers of rotations for reconstructing the 2D correlated geometric Brownian motion Eq. 22 in Example 3.4 when using the rotated squared $W_2$ distance loss function in Appendix D. The results are shown in Fig. S2.

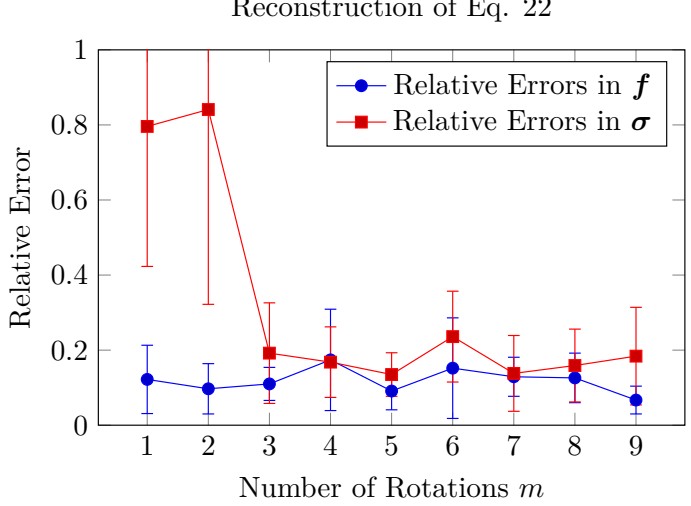

Figure S2: The relative errors in $\boldsymbol{f}$ defined in Eq. 19 and the relative errors in $\boldsymbol{\sigma}$ defined in Eq. 23 with respect to $m$, the number of rotations for the rotated squared $W_2$ loss function outlined in Appendix D when reconstructing the 2D correlated geometric Brownian motion Eq. 22 in Example 3.4.

From Fig. S2, there is a drastic improvement in the reconstruction accuracy of $\boldsymbol{\sigma}$ when the number of rotations $m$ is increased above 2. The matrix $\boldsymbol{\sigma}$ is not diagonal and thus the diffusion in $X_1, X_2$ are correlated, so rotations could help reconstruct the correlation of the diffusion term in Eq. 22.

On the other hand, the reconstruction accuracy of $\boldsymbol{f}$ does not improve significantly when the number of rotations is increased. Therefore, we carry out an additional numerical example of reconstructing the following 2D SDE

$$\mathrm{d}X_1(t) = \sum_{i=1}^{2} \mu_{1,i} X_i(t) + \sum_{i=1}^{2} \sigma_{1,i} X_i(t) \mathrm{d}B_i(t), \quad \mathrm{d}X_2(t) = \sum_{i=1}^{2} \mu_{2,i} X_i(t) + \sum_{i=1}^{2} \sigma_{2,i} X_i(t) \mathrm{d}B_i(t). \quad (51)$$

Eq. 51 is different from Eq. 22 in the drift term. To be more specific, both components in $\boldsymbol{f} := (\sum_{i=1}^{2} \mu_{1,i} X_i, \sum_{i=1}^{2} \mu_{2,i} X_i)$ depend on $X_1, X_2$. We apply the rotated squared $W_2$ distance with different numbers of rotations to reconstruct $\boldsymbol{f}$ and $\boldsymbol{\sigma} := [\sigma_{1,1} X_1, \sigma_{1,2} X_2; \sigma_{2,1} X_1, \sigma_{2,2} X_2]$ in Eq. 51. We set $(\mu_{1,1}, \mu_{1,2}, \mu_{2,1}, \mu_{2,2}) = (0.1, 0.05, 0.05, 0.2)$, and $\boldsymbol{\sigma} = [0.2X_1, -0.1X_2; -0.1X_1, 0.1X_2]$. The initial condition is set to be $(X_1(0), X_2(0)) = (1, 0.5)$ and we let $t \in [0, 2]$.

Reconstruction of Eq. 48

Figure S3: The relative errors in $\boldsymbol{f}$ defined in Eq. 19 and the relative errors in $\boldsymbol{\sigma}$ defined in Eq. 23 with respect to $m$, the number of rotations for the rotated squared $W_2$ loss function outlined in Appendix D when reconstructing the 2D correlated geometric Brownian motion Eq. 51. The parameters used in the neural network is the same as in Example 3.4.

From Fig. S3, we can see that even though both components of $\boldsymbol{f}$ depend on $X_1$ and $X_2$, there is no drastic improvement in the reconstruction error when the number of rotations $m$ is increased. However, the reconstruction error in $\boldsymbol{\sigma}$ becomes much smaller when $m$ is increased above 2. This indicates that the existence of correlations between diffusion terms confounds the reconstruction by the 2D squared $W_2$ distance defined in Appendix D (the 2D squared $W_2$ distance is the same as the $m = 1$ case of the rotated $W_2$ loss), while correlation in drift terms does not. The results in both Fig. S2 and Fig. S3 also show that our proposed rotated squared $W_2$ distance with $m \geq 3$ has the potential to resolve the correlation between $X_1$ and $X_2$ in both the drift and the diffusion terms. Overall, it is worth carrying out further analysis on how to efficiently apply the rotated squared $W_2$ distance to reconstruct general multi-dimensional SDEs.

## J ADDITIONAL DISCUSSION ON THE LOSS FUNCTION EQ. 17

Here, we make an additional comparison of using Eq. 16 and Eq. 17 as loss functions in Example 4. We set the number of training samples to be 128 and other hyperparameters for training to be the same as in the hyperparameters in Example 4 detailed in Table 1. First, we minimize Eq. 16 and record Eq. 16 and Eq. 17 over training epochs. Next, we minimize Eq. 17 and record Eq. 16 and Eq. 17 over training epochs. The results are shown in Fig. S4.

From Fig. S4 (a), we can see that when minimizing Eq. 16, Eq. 16 is almost $10^{0.5}$ times larger than Eq. 17. However, when minimizing Eq. 17, the values of Eq. 16 and Eq. 17 are close to each other (Fig. S4 (b)). In both cases, Eq. 17 converges to approximately $10^{-1}$. Interestingly, minimizing Eq. 17 leads to a smaller

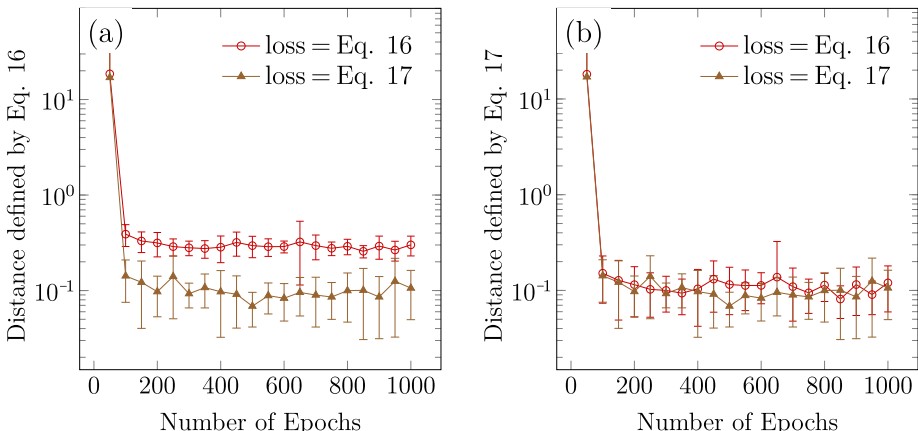

Figure S4: (a) The change in Eq. 16 and Eq. 17 when minimizing Eq. 16 over training epochs. (b) The change in Eq. 16 and Eq. 17 when minimizing Eq. 17 over training epochs.

discretized $W_2$ distance defined in Eq. 16. This again implies that minimizing Eq. 17 could be more effective than minimizing Eq. 16 in Example 3.4. Overall, more analysis on Eq. 17 is needed to understand its theoretical properties and compare the performances of minimizing Eq. 17 versus minimizing Eq. 16 from both theory and numerical aspects.

We let $\boldsymbol{\mu}_i, \hat{\boldsymbol{\mu}}_i$ be the two probability distributions on the space of continuous functions associated with $\boldsymbol{X}(t), t \in [t_i, t_{i+1})$ and $\hat{\boldsymbol{X}}(t), t \in [t_i, t_{i+1})$. We shall show that Eq. 17 is an approximation to the time-decoupled summation of squared $W_2$ distances $\sum_{i=1}^{N-1} W_2(\boldsymbol{\mu}_i, \hat{\boldsymbol{\mu}}_i)$ as $N \to \infty$. This result is similar to approximating $W_2^2(\mu, \hat{\mu})$ by Eq. 16 as $N \to \infty$ in Theorem 2.

We assume that the conditions in Theorem 2 hold true. By applying Theorem 2 with $N = 1$, the bound holds true for all $i = 1, 2, ..., N-1$

$$
\begin{aligned}
\inf_{\pi_i} \mathbb{E}_{\pi_i} [|\boldsymbol{X}(t_i) - \hat{\boldsymbol{X}}(t_i)|_2^2]^{\frac{1}{2}} \Delta t^{\frac{1}{2}} - \sqrt{(s+1)\Delta t} \big( (F_i \Delta t + \Sigma_i)^{\frac{1}{2}} + (\hat{F}_i \Delta t + \hat{\Sigma}_i)^{\frac{1}{2}} \big) \leq W_2(\boldsymbol{\mu}_i, \hat{\boldsymbol{\mu}}_i) \\
\leq \inf_{\pi_i} \mathbb{E}_{\pi_i} [|\boldsymbol{X}(t_i) - \hat{\boldsymbol{X}}(t_i)|_2^2]^{\frac{1}{2}} \Delta t^{\frac{1}{2}} + \sqrt{(s+1)\Delta t} \big( (F_i \Delta t + \Sigma_i)^{\frac{1}{2}} + (\hat{F}_i \Delta t + \hat{\Sigma}_i)^{\frac{1}{2}} \big).
\end{aligned}
\tag{52}
$$

In Eq. 52,

$$
\begin{aligned}
F_i &:= \mathbb{E}\Big[ \int_{t_i}^{t_{i+1}} \sum_{i=1}^{d} f_i^2(\boldsymbol{X}(t), t) \mathrm{d}t \Big] < \infty, \ \Sigma_i := \mathbb{E}\Big[ \int_{t_i}^{t_{i+1}} \sum_{\ell=1}^{d} \sum_{j=1}^{s} \sigma_{i,j}^2(\boldsymbol{X}(t), t) \mathrm{d}t \Big] < \infty, \\
\hat{F}_i &:= \mathbb{E}\Big[ \int_{t_i}^{t_{i+1}} \sum_{i=1}^{d} \hat{f}_i^2(\hat{\boldsymbol{X}}(t), t) \mathrm{d}t \Big] < \infty, \ \hat{\Sigma}_i := \mathbb{E}\Big[ \int_{t_i}^{t_{i+1}} \sum_{\ell=1}^{d} \sum_{j=1}^{s} \hat{\sigma}_{i,j}^2(\hat{\boldsymbol{X}}(t), t) \mathrm{d}t \Big] < \infty,
\end{aligned}
\tag{53}
$$

owing to the fact that

$$
\sum_{i=0}^{N-1} F_i = F < \infty, \ \sum_{i=0}^{N-1} \hat{F}_i = \hat{F} < \infty, \ \sum_{i=0}^{N-1} \hat{\Sigma}_i = \Sigma < \infty, \ \sum_{i=0}^{N-1} \hat{\Sigma}_i = \hat{\Sigma} < \infty,
\tag{54}
$$

where $F, \hat{F}, \Sigma, \hat{\Sigma}$ are defined in Eq. 14. Taking the square of the inequality 52, we have

$$W_2^2(\boldsymbol{\mu}_i, \hat{\boldsymbol{\mu}}_i) \leq \inf_{\pi_i} \mathbb{E}_{\pi_i}[|\boldsymbol{X}(t_i) - \hat{\boldsymbol{X}}(t_i)|_2^2]\Delta t + 2\inf_{\pi_i}\mathbb{E}_{\pi_i}[|\boldsymbol{X}(t_i) - \hat{\boldsymbol{X}}(t_i)|_2^2]^{\frac{1}{2}}$$

$$\times \sqrt{s+1}\Delta t\big((F_i\Delta t + \Sigma_i)^{\frac{1}{2}} + (\hat{F}\Delta t + \hat{\Sigma}_i)^{\frac{1}{2}}\big) + 2(s+1)\Delta t(F_i\Delta t + \Sigma_i + \hat{F}_i\Delta t + \hat{\Sigma}_i),$$

$$W_2^2(\boldsymbol{\mu}_i, \hat{\boldsymbol{\mu}}_i) \geq \inf_{\pi_i} \mathbb{E}_{\pi_i}[|\boldsymbol{X}(t_i) - \hat{\boldsymbol{X}}(t_i)|_2^2]\Delta t - 2W_2(\boldsymbol{\mu}_i, \hat{\boldsymbol{\mu}}_i)\sqrt{(s+1)\Delta t}\big((F_i\Delta t + \Sigma_i)^{\frac{1}{2}} + (\hat{F}\Delta t + \hat{\Sigma}_i)^{\frac{1}{2}}\big)$$

$$- 2(s+1)\Delta t(F_i\Delta t + \Sigma_i + \hat{F}_i\Delta t + \hat{\Sigma}_i) \tag{55}$$

Specifically, if there is a uniform bound $M > 0$ such that

$$\inf_{\pi_i} \mathbb{E}_{\pi_i}[|\boldsymbol{X}(t_i) - \hat{\boldsymbol{X}}(t_i)|_2^2]^{\frac{1}{2}} < M, \forall i = 1, ..., N-1, \tag{56}$$

then

$$W_2(\boldsymbol{\mu}_i, \hat{\boldsymbol{\mu}}_i) \leq \Delta t^{\frac{1}{2}}\Big(M + \sqrt{s+1}\big((FT + \Sigma)^{\frac{1}{2}} + (\hat{F}T + \hat{\Sigma})^{\frac{1}{2}}\big)\Big) := \tilde{M}\Delta t^{\frac{1}{2}}, \ \tilde{M} < \infty \tag{57}$$

Summing over $i = 1, ..., N-1$ for both inequalities in Eq. 55 and noticing that $\Delta t = \frac{T}{N}$, we conclude

$$\sum_{i=1}^{N-1} W_2^2(\boldsymbol{\mu}_i, \hat{\boldsymbol{\mu}}_i) \leq \sum_{i=1}^{N-1} \inf_{\pi_i}\mathbb{E}_{\pi_i}[|\boldsymbol{X}(t_i) - \hat{\boldsymbol{X}}(t_i)|_2^2]\Delta t$$

$$+ 2M\sum_{i=1}^{N-1}\sqrt{s+1}\Delta t\big((F_i\Delta t + \Sigma_i)^{\frac{1}{2}} + (\hat{F}_i\Delta t + \hat{\Sigma}_i)^{\frac{1}{2}}\big) + 2(s+1)\Delta t(F\Delta t + \Sigma + \hat{F}\Delta t + \hat{\Sigma}),$$

$$\leq \sum_{i=1}^{N-1} \inf_{\pi_i}\mathbb{E}_{\pi_i}[|\boldsymbol{X}(t_i) - \hat{\boldsymbol{X}}(t_i)|_2^2]\Delta t + 2(s+1)\Delta t(F\Delta t + \Sigma + \hat{F}\Delta t + \hat{\Sigma})$$

$$+ M\sqrt{s+1}\Delta t^{\frac{1}{2}}\big((F + \hat{F} + 2T)\Delta t^{\frac{1}{2}} + \Sigma + \hat{\Sigma} + 2T\big) \tag{58}$$

and

$$\sum_{i=1}^{N-1} W_2^2(\boldsymbol{\mu}_i, \hat{\boldsymbol{\mu}}_i) \geq \sum_{i=1}^{N-1} \inf_{\pi_i}\mathbb{E}_{\pi_i}[|\boldsymbol{X}(t_i) - \hat{\boldsymbol{X}}(t_i)|_2^2]\Delta t$$

$$- 2\tilde{M}\sum_{i=1}^{N-1}\sqrt{s+1}\Delta t\big((F_i\Delta t + \Sigma_i)^{\frac{1}{2}} + (\hat{F}_i\Delta t + \hat{\Sigma}_i)^{\frac{1}{2}}\big) - 2(s+1)\Delta t(F\Delta t + \Sigma + \hat{F}\Delta t + \hat{\Sigma}),$$

$$\geq \sum_{i=1}^{N-1} \inf_{\pi_i}\mathbb{E}_{\pi_i}[|\boldsymbol{X}(t_i) - \hat{\boldsymbol{X}}(t_i)|_2^2]\Delta t - 2(s+1)\Delta t(F\Delta t + \Sigma + \hat{F}\Delta t + \hat{\Sigma})$$

$$- \tilde{M}\sqrt{s+1}\Delta t^{\frac{1}{2}}\big((F + \hat{F} + 2T)\Delta t^{\frac{1}{2}} + \Sigma + \hat{\Sigma} + 2T\big) \tag{59}$$

Eqs. 58 and 59 indicate that as $N \to \infty$,

$$\sum_{i=1}^{N-1} \inf_{\pi_i}\mathbb{E}_{\pi_i}[|\boldsymbol{X}(t_i) - \hat{\boldsymbol{X}}(t_i)|_2^2]\Delta t - \sum_{i=1}^{N-1} W_2^2(\boldsymbol{\mu}_i, \hat{\boldsymbol{\mu}}_i) \to 0, \tag{60}$$

Specifically, if the limit

$$\lim_{N \to \infty} \sum_{i=1}^{N-1} W_2^2(\boldsymbol{\mu}_i, \hat{\boldsymbol{\mu}}_i) \tag{61}$$

exists, denoted by $\tilde{W}_2^2(\mu, \hat{\mu})$, then we conclude that $\tilde{W}_2^2(\mu, \hat{\mu})$ can be approximated by $\sum_{i=1}^{N-1} \inf_{\pi_i} \mathbb{E}_{\pi_i}[|\boldsymbol{X}(t_i) - \hat{\boldsymbol{X}}(t_i)|_2^2] \Delta t$ (the left hand side of Eq. 17) as $\Delta t \to 0$, which is similar to the effectiveness of approximating $W_2(\mu, \hat{\mu})$ by $W_2(\mu_N, \hat{\mu}_N)$ proved in Theorem 2. Furthermore, from Eq. 17, we can deduce that

$$\tilde{W}_2^2(\mu, \hat{\mu}) = \lim_{N \to \infty} \sum_{i=1}^{N-1} \inf_{\pi_i} \mathbb{E}_{\pi_i}[|\boldsymbol{X}(t_i) - \hat{\boldsymbol{X}}(t_i)|_2^2] \Delta t \le \lim_{N \to \infty} W_2^2(\mu_N, \hat{\mu}_N) = W_2^2(\mu, \hat{\mu}). \tag{62}$$

Therefore, the upper bound of $W_2^2(\mu, \hat{\mu})$ in Theorem 1 is also an upper bound of $\tilde{W}_2^2(\mu, \hat{\mu})$, *i.e.*, to reconstruct a 1D SDE, minimizing $\tilde{W}_2^2(\mu, \hat{\mu})$ is necessary to have small $f - \hat{f}$ and $\sigma - \hat{\sigma}$ for $f, \hat{f}, \sigma, \hat{\sigma}$ defined in Eqs. 1 and 2. Thus, minimizing the loss function defined in Eq. 17, when $\Delta t$ is sufficiently small, is also necessary for minimizing the errors in $\hat{f}$ and $\hat{\sigma}$ of the reconstructed SDE Eq. 2. The analysis above provides intuitive and heuristic understanding of the loss function Eq. 17. Further study on the existence of $\tilde{W}(\mu, \hat{\mu})$ deserves attention.

## K   APPLICATION IN RECONSTRUCTING A 5D SDE FEATURING CIRCADIAN CLOCKS

Finally, as an application of our SDE reconstruction approach in biology, we reconstruct a five-dimensional SDE model which is derived by adding Brownian noise to five coupled ODEs describing a periodic circadian clock (Goldbeter, 1995).

$$\begin{aligned}
\mathrm{d}M &= \left(v_s \frac{K_I^4}{K_I^4 + P_N^4} - v_m \frac{M}{K_m + M}\right)\mathrm{d}t + 0.1M\mathrm{d}B_t^1, \\
\mathrm{d}P_0 &= \left(k_s M - v_1 \frac{P_0}{K_1 + P_0} + v_2 \frac{P_1}{K_2 + P_1}\right)\mathrm{d}t + 0.05P_0\mathrm{d}B_t^2, \\
\mathrm{d}P_1 &= \left(v_1 \frac{P_0}{K_1 + P_0} - v_2 \frac{P_1}{K_2 + P_1} - v_3 \frac{P_1}{K_3 + P_1} + v_4 \frac{P_2}{K_4 + P_2}\right)\mathrm{d}t + 0.1\mathrm{d}B_t^3, \\
\mathrm{d}P_2 &= \left(v_3 \frac{P_1}{K_3 + P_1} - v_4 \frac{P_2}{K_4 + P_2} - k_1 P_2 + k_2 P_N - v_d \frac{P_2}{K_d + P_2}\right)\mathrm{d}t, \\
\mathrm{d}P_N &= \left(k_1 P_2 - k_2 P_N - v_n \frac{P_N}{K_n + P_N}\right)\mathrm{d}t + 0.01\mathrm{d}B_t^4, \ \ t \in [0, 50].
\end{aligned} \tag{63}$$

In Eq. 63, $M$ describes the concentration of mRNA, $P_0$ is the concentration of native protein, $P_1, P_2$ represent concentrations of two different forms of phosphorylated protein, and $P_N$ quantifies the concentration of nuclear protein. The parameters $K_1 = 2\mu\text{mol}, K_2 = 2\mu\text{mol}, K_3 = 2\mu\text{mol}, K_4 = 2\mu\text{mol}, K_n, K_I = 1\mu\text{mol}$, and $K_m = 0.5\mu\text{mol}$ are concentration parameters associated with Michaelis-Menten kinetics. Reaction rates are represented by $v_s = 0.76\mu\text{mol}/h, v_1 = 3.2\mu\text{mol}/h, v_2 = 1.58\mu\text{mol}/h, v_3 = 5\mu\text{mol}/h, v_4 = 2.5\mu\text{mol}/h, v_m = 0.65\mu\text{mol}/h, v_d = 0.95\mu\text{mol}/h$, and $k_s = 0.38h^{-1}, k_1 = 1.9h^{-1}, k_2 = 1.3h^{-1}$. The dynamics involve four independent Brownian noises described by $\mathrm{d}B_t^i, i = 1, 2, 3, 4$.

We plot the ground truth trajectories and the trajectories generated using our $W_2$-distance SDE reconstruction method in Fig. S5. The training details are given in Table 1. For simplicity, we plot the mRNA concentration $M$, the naive protein concentration $P_0$, and the nuclear protein concentration $P_N$, which all display periodic fluctuations over time. The reconstructed trajectories by our Wasserstein-distance SDE approach can accurately reproduce the noisy periodic changes in the mRNA and protein concentrations.

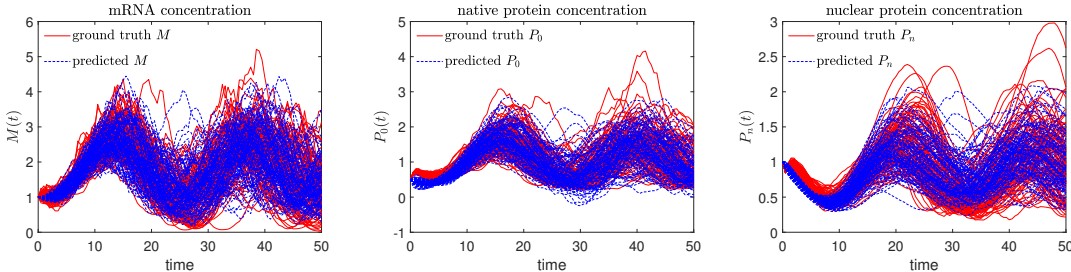

Figure S5: The reconstructed trajectories using our Wasserstein-distance SDE reconstruction approach compared to the ground truth trajectories obtained by simulating Eqs. equation 63. For simplicity, we plot the ground truth and reconstructed trajectories of the concentrations of mRNA, native protein, and nuclear protein. The initial condition is set as $(M(0), P_0(0), P_1(0), P_2(0), P_N(0)) = (1, 0.5, 2, 0, 1)$ (unit: $\mu$mol) for all trajectories.

