# OpenReview forum: "A Wasserstein-2 Distance for Efficient Reconstruction of Stochastic Differential Equations"
_ICLR.cc/2024/Conference — Submitted to ICLR 2024_

### Official Review · Reviewer_2Kjt · 2023-10-29

**Soundness:** 2 fair
**Presentation:** 2 fair
**Contribution:** 2 fair
**Rating:** 3
**Confidence:** 2

**Summary:**

The paper derives Wasserstein distance between two probability distributions associated with two stochastic differential equations.

**Strengths:**

The paper tries to learn a SDE by bounding the W2 distance.

**Weaknesses:**

There are no generalization bounds.

**Questions:**

Can we prove some finite sample generalization bounds?

---

> ### Author Response · Authors · 2023-11-16
> **Rebuttal responses to Reviewer 2Kjt**
>
> Thank you for your careful reading and valuable suggestions.  We
> believe that there could some misunderstanding on the ``generalization
> bound".
>
> To summarize, to the best of our knowledge, there are very few
> published methods for general SDE reconstruction.  We propose the
> first general framework that can be readily used to reconstruct
> general SDEs based on the Wasserstein distance. It provides a simple
> and efficient (compared to recent Wasserstein generative adversarial
> method and MMD loss method) way to reconstruct SDEs.  We applied our
> new Wasserstein-distance-based SDE reconstruction method noisy
> trajectories that arise in many biological and biophysical
> settings. We show some of our preliminary results here
> https://drive.google.com/file/d/1HW5CBjx7g37DFx3s3A-sNDK0aoG6zqbT/view?usp=drive_link.
> We will submit more extensive results on arXiv soon.
>
> Given the novelty and potentially broad applicability of our method
> towards uncertainty quantification and analyzing time series data, we
> hope you will kindly consider reevaluating the revised manuscript and
> update your score in light of our responses:
>
> Q1 \& W1. Thank you for your question, and we appreciate the opportunity to
> clarify our approach in addressing your question on ``generalization
> bounds".
>
> Our Theorem 2 provides a time-discretization error bound for the difference
> between the Wasserstein
> distance of solutions to two distinct SDEs given the same initial condition and the Wasserstein distance
> of finite-time-point observations of the solutions to those two SDEs.
>
> The
> stochastic nature of the Stochastic Differential Equation (SDE) we aim
> to reconstruct inherently complicates the straightforward extension of
> "generalization bounds" defined for deterministic systems. Thus, in Examples
> 2, 3, and 4, we evaluate the performance of SDE reconstruction using
> the metrics $|f-\hat{f}|$ and $|\sigma-\hat{\sigma}|$, representing
> the errors in reconstructing the drift and diffusion terms of the SDE,
> respectively. This choice aligns with our method's objective of
> reconstructing general unknown SDEs from data.
>
>
> It would be nice to obtain ``finite sample generalization
> bounds". Even though a theoretical derivation of such bounds is quite nontrivial
> challenging, we have conducted a sensitivity analysis in Examples 2,
> 3, and 4 to investigate how the number of samples impacts the
> reconstruction of $f$ and $\sigma$ (refer to Figs. 2, 3, and 4). Our preliminary results
> indicate that more samples lead to better accuracy and smaller $|f-\hat{f}|$ and $|\sigma-\hat{\sigma}|$.
>
> Analyzing the error bound between the empirical Wasserstein distance
> and the genuine Wasserstein distance is complicated and depends on
> both the number of samples and the dimensionality of the SDE. This is
> beyond the scope of our paper, as our focus was to introduce and
> develop a novel and practical (see our examples and additional applications)
> Wasserstein-distance-based SDE reconstruction method as well as
> provide insight into its efficacy.
>
>
> In theory, there is also a theorem for analyzing the errors between the
> empirical Wasserstein distance with finite samples and the genuine
> Wasserstein distance, see, \textit{e.g.}
>
> Ref [1]. Fournier, Nicolas, and Arnaud Guillin, On the rate of
> convergence in Wasserstein distance of the empirical measure,
> $\textbf{Probability theory and related fields}$, 162(3-4): 707-738,
> (2015).
>
> Currently, our ongoing efforts involve both theoretical analysis of
> our Wasserstein-distance SDE reconstruction method and practical
> applications in reconstructing multiple SDEs with noisy dynamics in
> biology. We have
> cited Ref [1] in our Example 4 to clearly state that more analysis is
> needed on the number of samples and dimensionality in our future work.
>
> We hope that these clarifications effectively address your concerns
> and we are open to further discussion or inquiries about our new
> Wasserstein-distance-based loss function and SDE inference approach.

---

> ### Author Response · Authors · 2023-11-21
>
> Dear Reviewer 2Kjt,
>
> We are grateful for the time and expertise you dedicated to reviewing our paper.
> In light of the clarifications we have made in response to your comments, we would be immensely appreciative if you could take a moment to review our responses. We hope that our clarifications and responses have satisfactorily addressed your questions and concerns and we would greatly appreciate it if our responses could warrant your reevaluation of our work. Thank you once again for your invaluable contribution and for sharing your expertise with us!
>
> With sincere gratitude,
>
> Authors

---

### Official Review · Reviewer_ioMy · 2023-11-01

**Soundness:** 2 fair
**Presentation:** 3 good
**Contribution:** 2 fair
**Rating:** 6
**Confidence:** 3

**Summary:**

The paper analyzes the Wasserstein 2 distance (WD) between solutions of two stochastic differential equations of form $d X(t) = f(X(t),t) d t + \sigma(X(t),t) d B(t)$ as distributions over continuous function space. The main result shows that under certain mild assumptions, the WD is upper bounded by differences over $f$ and $\sigma$, which establishes the relation between estimation of $f,\sigma$ and the resulting reconstruction. Furthermore, when only a finite dimensional distribution is observed, the WD between the finite projections and the original WD are shown to be close to each other. An approximator of the finite projected WD is also proposed, using which various examples are implemented.

**Strengths:**

The paper is overall well written and presented, and the the ideas are original to the knowledge of the reviewer. Some strengths:
1. The bound in Theorem 1 seems interesting and novel, which establishes how the solution to SDE evolves as the parameters change. This provides evidence that WD is a valid loss function for SDE reconstruction. Moreover, an analysis of WD over Banach space (continuous function space) seems interesting.
2. The proposed finitely projected WD loss seems simple to compute, and the examples illustrate that it is suitable for various reconstruction tasks.

**Weaknesses:**

1. Theorem 1 is an interesting bound, but it is unclear how this gives a full characterization of what happens when using WD as a loss for reconstruction. In fact, a reverse of the inequality would also be needed to show that when minimizing the WD, the functions $f,\sigma$ are both well approximated, as it is usually the WD that we can observe, rather than gaps for $f,\sigma$.
2. A major concern is the tightness of equation (17). As a simple example, if $f,\sigma$ are both only function of $t$ not $x$, then the finite projections will be jointly Gaussian, the WD between which is not likely to be just sum of WD between marginal. Thus it is hard to assess how good the approximation is, even if it seems to work well in the provided examples.

**Questions:**

Please see above (section Weaknesses) for details. The major question is how tight the approximation (17) is, as it was extensively used in the examples, and is claimed to be an approximation of the original WD. It is hard to assess how well the reconstruction is from the current contents of the paper, as it does not seem to properly 'interpolate' between discrete time steps if only local information is used, and continuity of the construction is provided by the continuity of the neural network estimators. Alternatively the paper can also provide theoretical guarantees specifically for this loss.

---

> ### Author Response · Authors · 2023-11-16
> **Rebuttal responses to Reviewer ioMy part I**
>
> Thank you for your careful reading and valuable suggestions.  We have
> revised our manuscript based on your comments.
>
> To summarize, to the best of our knowledge, there are very few
> published methods for general SDE reconstruction.  We propose the
> first general framework that can be readily used to reconstruct
> general SDEs based on the Wasserstein distance. It provides a simple
> and efficient (compared to recent Wasserstein generative adversarial
> method and MMD loss method) way to reconstruct SDEs.  We applied our
> new Wasserstein-distance-based SDE reconstruction method noisy
> trajectories that arise in many biological and biophysical
> settings. We show some of our preliminary results here
> https://drive.google.com/file/d/1HW5CBjx7g37DFx3s3A-sNDK0aoG6zqbT/view?usp=drive_link.
> We will submit more extensive results on arXiv soon.
>
> Given the novelty and potentially broad applicability of our method
> towards uncertainty quantification and analyzing time series data, we
> hope you will kindly consider reevaluating the revised manuscript and
> update your score in light of our responses:
>
> W1. Thank you for your insightful
> observation. We would like to first clarify how Theorem 1 contributes
> to a comprehensive understanding of how to use the Wasserstein
> distance (WD) as a loss for SDE reconstruction. Theorem 1 provides an
> interesting upper bound for the Wasserstein distance concerning the
> discrepancies $|f-\hat{f}|$ and $|\sigma-\hat{\sigma}|$. This upper
> bound points to the necessity of minimizing the WD for the successful
> reconstruction of both $f$ and $\sigma$. We have explicitly
> highlighted this point on Page 3.
>
> It is nontrivial to show a lower bound on the WD
> as a function of the errors in the drift and diffusion terms
> $f-\hat{f}$ and $\sigma-\hat{\sigma}$. The challenge lies in
> the definition of the Wasserstein distance (Definition 2.1), which
> involves taking the infimum over all admissible couplings $\pi(\mu,
> \hat{\mu})$. Finding a lower bound for WD would probably require
> constructing a specific coupling $\pi^*(\mu, \hat{\mu})$ to establish
> $W_2(\mu, \hat{\mu})\approx \mbox{E}_{(\boldsymbol{X}, \hat{\boldsymbol{X}})\sim
>   \pi^*(\boldsymbol{X}, \hat{\boldsymbol{X}})}[\lVert\boldsymbol{X}-\hat{\boldsymbol{X}}\rVert^2]$, which can
> be analytically challenging.  In essence, the squared Wasserstein
> distance $W_2^2$ serves as a measure of "error". Typically, we can
> only derive upper bounds on this error.
>
> Most of the literature analyzes Wasserstein distances applied to SDEs,
> such as
>
> Ref [1]. Wang Jian, $L^p$-Wasserstein distance for stochastic differential
> equations driven by L\'evy processes, $\textbf{Bernoulli}$, (2016):
> 1598-1616.
>
> Ref [2]. Sanz-Serna, Jesus Maria, and Konstantinos C. Zygalakis,
> Wasserstein distance estimates for the distributions of numerical
> approximations to ergodic stochastic differential equations,
> $\textbf{The Journal of Machine Learning Research}$, 22.1 (2021):
> 11006-11042.
>
> These works obtained only an upper bound for the Wasserstein distance
> and found no lower bounds.  To the best of our knowledge, we do not
> know any literature on how to derive effective lower bounds for the WD
> unless one imposes some restrictions on the coupling $\pi$, such as in
>
> Ref [3]. M\'emoli, Facundo, A spectral notion of Gromov–Wasserstein
> distance and related method, $\textbf{Applied and Computational
>   Harmonic Analysis}$, 30.3: 363-401, (2011).
>
> and other papers that study some regularized Wasserstein
> distances.
>
> While the feasibility of establishing a lower bound for WD remains
> uncertain, our numerical examples demonstrate the efficacy of training
> with WD. Undoubtedly, the exploration of deriving a lower bound for WD
> in the context of solutions to two SDEs warrants further
> investigation. We are committed to addressing this gap in our future
> work and will make our findings public once we find meaningful
> results.

---

> ### Author Response · Authors · 2023-11-16
> **Rebuttal responses to Reviewer ioMy part II**
>
> W2. We appreciate your thoughtful
> consideration of Eq. (17) and your concern regarding its tightness.
>
> Eq. (17) is an approximation to Eq. (16), and it is helpful to
> evaluate how it approximates Eq. (16) in general cases.  One rationale
> behind employing Eq. (17) in our examples is its advantageous
> simplification for univariate SDEs. In such cases, Eq. (17) reduces to
> Eq. (3), allowing for analytical calculations and a clear statistical
> interpretation. Specifically, Eq. (17) aims to match the distribution
> of reconstructed trajectories with the ground truth trajectories at
> different time points.
>
> In Example 4, we also directly minimized Eq. (16) (the blue line) as
> the loss function. Our result shows that minimizing Eq. (16) performs
> worse than minimizing Eq. (17). To provide further insight, we
> incorporated a visualization of the loss functions concerning training
> epochs in Example 4, demonstrating the superior numerical efficiency
> of minimizing Eq. (17) relative to minimizing Eq. (16).
>
> In theory, numerically calculating Eq. (16) could be less accurate
> especially when the number of samples is limited, as the bound of the
> discrepancy between the empirical Wasserstein distance and the genuine
> Wasserstein distance deteriorates when the dimensionality increases.
> In using Eq. (16), the dimensionality is $Nd$, where $N$ is the number
> of timesteps and $d$ is the dimensionality of the SDE. See,
> \textit{e.g.},
>
> Ref [4]. Nicolas Fournier and Arnaud Guillin, On the rate of convergence in
> Wasserstein distance of the empirical measure, $\textbf{Probability
>   Theory and Related Fields}$, 162.3-4: 707-738, (2015).
>
> A comprehensive comparison between
> Eq. (16) and the sum of Wasserstein distances between marginals
> (Eq. 17) requires further discussion, especially when dealing with
> multidimensional SDEs, which we outlined in Example 4. This intricate analysis goes beyond the scope
> of our current manuscript, which primarily introduces the novel
> WD-based SDE reconstruction method and provides preliminary insight
> into its efficacy.
>
> As applications, we have successfully applied Eq. (17) as a loss
> function for reconstructing multiple multivariate SDEs.  Currently, we
> are working on analyzing WD for multidimensional SDEs and exploring
> whether there are more suitable loss functions than Eq. (17).
>
> We hope that our responses can satisfactorily address your questions
> and concerns and that you may better appreciate the new,
>   practical, and efficient Wasserstein-distance-based SDE
>   reconstruction approach that we developed.

---

> > ### Comment · Reviewer_ioMy · 2023-11-20
> > **Thanks for the response**
> >
> > I thank the authors for the response, which addresses most of my questions. Regarding (17) I'm still not fully convinced that the proposed method is a principled substitute of (16), as it seems all the causality across different time are lost, even if (17) performs better empirically. I will thus still leave my score unchanged.

---

> > > ### Author Response · Authors · 2023-11-21
> > > **Thank you for your suggestion & kindly request for reevaluation**
> > >
> > > Thank you for your valuable comments.
> > >
> > > We wish to bring to your attention that both Eq. (16) and Eq. (17) are novel Wasserstein-distance-based loss functions for SDE reconstruction proposed by us, which provide two potential loss functions for reconstructing SDEs. Empirically, Eq. (17) performs better than Eq. (16). In our last response, we have also explained why numerically calculating Eq. (16) might not be so good due to the high dimensionality when coupling all time points. Furthermore, in our revised manuscript we also wrote in Example 4 that further discussion is needed to devise an optimal Wasserstein-distance-based loss function.
> > > Given the novelty and broad applications of our method in uncertainty quantification and general SDE reconstruction in biology, physics, and other related fields (as shown in https://drive.google.com/file/d/1HW5CBjx7g37DFx3s3A-sNDK0aoG6zqbT/view), we hope that our manuscript warrants your reevaluation.

---

> > > > ### Comment · Reviewer_ioMy · 2023-11-21
> > > > **Thanks for the response**
> > > >
> > > > I thank again the authors for the response.
> > > >
> > > > Here are some clarification of my previous question: I acknowledge that the proposed (17) is rather easy to implement and performs well as illustrated. However, it seems the theoretical guarantees (e.g. Thm 2) are all for (16). The $\approx$ in (17) suggests that they are strongly related, but there are plenty of examples where (16) and (17) differs substantially. Thus it seems there's a noticeable gap between the theoretical guarantee part of the paper and the implementation part. The reviewer expects at least some direct justification is provided for (17), rather than simply stating "If one disregards the temporal correlations of values at different times within a single trajectory...". For the current form of the paper, it seems the soundness of (17) is only supported heuristically.
> > > >
> > > > The reviewer understands that this is additional work, so here are some alternative improvements: either to provide a bound or discussion for why the "$\approx$" in (17) makes sense, or simply stating (17) as a practically motivated quantity that is inspired by (16). In either case the reviewer suggests to add discussion on tightness or gap on the approximation argument $\approx$ in (17).
> > > >
> > > > One side note: reconstructing based on marginals (17) have appeared in a related context of Wasserstein splines [1,2,3]. Introducing this to inferring SDE coefficients indeed seems interesting, so the reviewer increases the rating to 6. (Personally the reviewer thinks it's possible to extend Thm 2 to (17) by smoothness assumptions of $f,\sigma$, which would greatly strengthen the theoretical merit of the paper.)
> > > >
> > > > [1] Chen, Yongxin, Giovanni Conforti, and Tryphon T. Georgiou. "Measure-valued spline curves: An optimal transport viewpoint." SIAM Journal on Mathematical Analysis 50.6 (2018): 5947-5968.
> > > > [2] Benamou, Jean-David, Thomas O. Gallouët, and François-Xavier Vialard. "Second-order models for optimal transport and cubic splines on the Wasserstein space." Foundations of Computational Mathematics 19 (2019): 1113-1143.
> > > > [3] Chewi, Sinho, et al. "Fast and smooth interpolation on Wasserstein space." International Conference on Artificial Intelligence and Statistics. PMLR, 2021.

---

> ### Author Response · Authors · 2023-11-22
> **Thanks for your suggestions! & Revisions based on your advice**
>
> Dear Reviewer ioMy,
>
> We really appreciate your feedback and your reevaluation!
> We are grateful for your constructive suggestions. Our previous statement that ``Eq. (17) is an approximation to Eq. (16)" was not very precise. Actually, Eq. (17) is smaller than or equal to Eq. (16) due to a relaxation on the constraint of the coupling \pi(\mu, \hat{\mu}). We have made this point clear and revised the manuscript accordingly (Page 5). Since Eq. (17)\leq Eq. (16), minimizing Eq. (17) is also necessary to have a small reconstruction error $f-\hat{f}$ and $\sigma-\hat{\sigma}$ owing to Theorems 1 & 2, and we have also made this point clear in the manuscript (Page 5).
>
> We have added an additional section (Appendix J) to carry out an additional numerical experiment in response to your question on the gap between Eq. (16) and Eq. (17). Interestingly, when minimizing Eq.(16), the gap between Eq. (16) and Eq. (17) is relatively large. However, when minimizing Eq. (17), the gap is small and both Eq.(16) and Eq.(17) are kept small. We believe that this is worth further investigation.
>
> Furthermore, in Appendix J, we have extended Thm 2 to Eq. (17) and explicitly showed that Eq. (17) is an approximation to the summation of time-decoupled squared W2
> distances
> \begin{equation}
> \sum_{i=1}^{N-1}W_2^2(\mu_i, {\hat{\mu_i}})
> \end{equation}
>  in the $N\rightarrow \infty$ limit, where $\mu_i, \hat{\mu_i}$ are the two probability distributions associated with $\boldsymbol{X}(t), t\in[t_i, t_{i+1})$ and $\hat{\boldsymbol{X}}(t), t\in[t_i, t_{i+1})$. This is our preliminary result on the theoretical aspect of the loss function Eq. (17). Please see our updated manuscripts for those revisions.
>
> Your references on the Wasserstein splines are greatly appreciated, and we have cited Ref. [3] which seems to be related to our time-decoupled W2 loss. We are working on further analyzing the squared $W_2$ distance, and will post our findings on arXiv once we reach some interesting results. We believe that our proposed simple and efficient W2-distance-based SDE reconstruction method could become a potential benchmark method in this specific SDE reconstruction field. We would greatly appreciate it if you found our work interesting and potentially useful.
>
>
> Sincerely,
>
> Authors

---

### Official Review · Reviewer_fb1D · 2023-11-01

**Soundness:** 4 excellent
**Presentation:** 3 good
**Contribution:** 3 good
**Rating:** 6
**Confidence:** 1

**Summary:**

This paper delved into the poblem of reconstructing SDEs (using Neural Networks) with Wasserstein-2 loss ($W_2$). The authors introduced a new bound on the $W_2^2$ loss of 2 probability measures constructed on 2 random process. In addition, the authors also provided an estimation of the $W_2^2$ that can be carried out numerically. Finally, the authors conducted empirical experiments testing the perfomance of reconstructing SDEs using Neural Networks with this $W_2^2$ loss.

**Strengths:**

1. The paper is written clearly with detailed explanation.

2. The authors provided an upper bound on the $W_2^2$ distance that can be approximated numerically.

3. The empirical performance of the new loss out-performs traditional MSE and KL loss.

**Weaknesses:**

I am not very familiar with this topic. There seems to be no major flaw in this paper, but I think it would be better if there are some more detailed comparisons with prior works and other losses, as the authors only provided some intuitive comparison with MSE and KL divergence losses in Appendix B.

**Questions:**

1. What kind of noise can the $W_2^2$ loss deal with?

2. Can the authors provide some intuition about why $W_2^2$ loss is more robust to noise?

3. Is eq(10) purely of intellectual interest or thers are some methods to estimate this expectation? It seems Theorem 2 didn't use this upper bound to construct an estimation of the $W_2^2$ loss.

4. Finally, I am curious what method did the authors use to minimize the $W_2^2$ loss?

---

> ### Author Response · Authors · 2023-11-16
> **Rebuttal responses to Reviewer fb1D**
>
> Thank you for your careful reading and valuable suggestions.
>
> To summarize, to the best of our knowledge, there are very few
> published methods for general SDE reconstruction.  We propose the
> first general framework that can be readily used to reconstruct
> general SDEs based on the Wasserstein distance. It provides a simple
> and efficient way (compared to recent Wasserstein generative
> adversarial method and MMD loss method) to reconstruct SDEs.  We
> applied our new Wasserstein-distance-based SDE reconstruction method
> to noisy trajectories that arise in many biological and biophysical
> settings. We show some of our preliminary results here
> https://drive.google.com/file/d/1HW5CBjx7g37DFx3s3A-sNDK0aoG6zqbT/view?usp=drive_link.
> We will submit more extensive results on arXiv soon.
>
> Given the novelty and potentially broad applicability of our method
> towards uncertainty quantification and analyzing time series data, we
> hope you will kindly consider reevaluating the revised manuscript and
> update your score in light of our specific responses:
>
> W1. Indeed in Appendix B we explained in theory why the MSE and KL
> divergence is not as good as the Wasserstein distance for the purpose
> of SDE reconstruction. Moreover, in Examples 1 and 2, we numerically
> compared our Wasserstein-distance based SDE reconstruction approach
> with MSE, maximal log likelihood, and Mean$^2$+Var approaches. Our
> Wasserstein distance loss gives the best reconstructed trajectories
> (Example 1) and smallest errors $f-\hat{f}$ and $\sigma-\hat{\sigma}$
> (Example 2). In Examples 3 and 4, we compare our approach with
> recently proposed machine learning methods such as a Wasserstein
> Generative Adversarial Network (WGAN) approach and a Maximum Mean
> Discrepancy (MMD) loss. Once again, our Wasserstein distance method
> outperforms the WGAN and MMD approaches in both accuracy and
> efficiency.  We hope that these numerical experiments and comparisons
> with other methods convince you of the advantages of our new method.
>
> Q1. In principle, the $W_2^2$ loss can can
> be calculated for a stochastic process obeying many kinds of noise.
> For example, the Wasserstein distance between different
>   solutions associated with different initial conditions of a general L\'evy process has been computed
> , \textit{e.g.},.
>
> Ref [1]. Jian Wang, $L^p$-Wasserstein distance for stochastic differential equations driven by L\'evy processes, $\textbf{Bernoulli}$, 1598-1616, (2016).
>
> In our work, we minimize a Wasserstein loss to
> reconstruct the SDE from data, and restrict ourselves to Brownian
> noise processes.
>
>
> Q2. The
> robustness of the $W_2^2$ loss to noise can be attributed to its
> nature as a distribution-based loss function, as exemplified by
> Eq. (3). This type of loss is designed to quantify the difference
> between the targeted distribution and the approximated distribution,
> making it particularly well-suited for reconstructing the distribution
> of trajectories. Compared to the MSE, $W_2^2$ captures the variation and compared to the Mean$^2$+Var , $W_2^2$ is able to capture more features such as multiple peaks,
> as illustrated in Example 1.
>
> In Example 1, we also demonstrated the suitability of the $W_2^2$ loss for
> reconstructing distributions. An additional advantage lies in the
> flexibility of the $W_2^2$ loss. Unlike other commonly used loss
> functions, such as Mean$^2$+Var or maximal log likelihood, the $W_2^2$
> loss does not necessitate an assumption of a Gaussian
> distribution. This characteristic enhances its robustness to various
> types of noise, as it does not rely on specific distributional
> assumptions.
>
> Q3. Eq. (10) is for Theorem 1. It is
> a rather technical condition such that if satisfied, then
> the $W_2^2$ distance between the solutions to two SDEs converges
> uniformly to 0 as $\hat{f}\rightarrow f, \hat{\sigma}\rightarrow
> \sigma$, as we explain on Page 4.
>
> As a simple case, if $f, \sigma, \partial_x f, \partial_x \sigma$ are
> uniformly bounded, then Eq. (10) is also uniformly bounded, and all
> assumptions of Theorem 1 are satisfied.
>
> Q4. As described
> on Page 6, training details are given in Appendix E. For all examples,
> the AdamW stochastic optimization method is used for minimizing the
> $W_2^2$ loss. The code implementation is available in the
> supplementary files provided with the manuscript.
>
> We hope that our responses satisfactorily address your questions and
> concerns and that you will better appreciate the new
> Wasserstein-distance-based SDE reconstruction method we have
> developed.

---

> > ### Comment · Reviewer_fb1D · 2023-11-21
> >
> > I thank the authors for their detailed response. I would like to keep my ratings unchanged.

---

> ### Author Response · Authors · 2023-11-21
>
> Dear Reviewer fb1D,
>
> We are grateful for the time and expertise you dedicated to reviewing our paper.
> In light of the changes and clarifications we have made in response to your comments, we would be immensely appreciative if you could take a moment to review our responses. Your confirmation on whether our responses have adequately addressed your concerns would be greatly valued. We hope that our clarifications and responses have satisfactorily addressed your questions and concerns and we would greatly appreciate it if our responses could warrant your reevaluation of our work.
> Thank you once again for your invaluable contribution and for sharing your expertise with us!
>
> With sincere gratitude,
>
> Authors

---

> ### Author Response · Authors · 2023-11-22
>
> Dear Reviewer fb1D,
>
> We really appreciate your time in reading our responses and your feedback!
>
> Extracting SDEs from noisy time series data has been an important topic in uncertainty quantification for biological and physical applications. However, there exist very few methods for SDE reconstruction. Using the W2 distance as a loss function to reconstruct SDEs is quite easy to implement and easy to understand. Furthermore, it outperforms traditional methods and recent machine learning methods we have known so far in both efficiency and accuracy. Thus, we believe that our proposed simple W2-distance-based SDE reconstruction could become a potential benchmark method in this specific SDE reconstruction field. We would greatly appreciate it if you found our work interesting and potentially useful.
>
> Sincerely,
>
> Authors

---

### Official Review · Reviewer_NdWD · 2023-11-01

**Soundness:** 2 fair
**Presentation:** 2 fair
**Contribution:** 2 fair
**Rating:** 6
**Confidence:** 4

**Summary:**

Optimal transport (OT) metrics, aka Wasserstein distances, are ubiquitous in comparing probability measures since they capture the underlying geometric of the objects at hand. This paper introduces the Wasserstein distance between two probability distributions associated with two solutions of stochastic differential equations (SDE). The author(s) investigate the squared Wasserstein distance to reconstruct the drift and the diffusion components of an SDE through a neural SDE.

**Strengths:**

- Investigating the 2-Wasserstein distance between two probability distributions associated with two solutions.
- Theoretical guarantees of the 2-Wasserstein distance.
- Reconstruction of the drift and the diffusions component of SDE through a neural SDE where the 2-Wasserstein distance serves as the loss training.
- Numerical experiments of univariate and bivariate SDE for CIR, OU, and 2D geometric Brownian models.

**Weaknesses:**

- I suggest that the result in Theorem 1 should be highlighted in the case of **univariate** SDE ($d=1$), since in Definition 1, the Wasserstein distance is defined for $d$-dimensional processes.
- In Remark 1, the author(s) mentioned that the upper bound could be generalized for $d$-dimensional case under mild assumptions. I think it is important to write this result and get some consistent presentation in the manuscript since Theorem 2 is valid for multivariate case.
- Again with Remark 1, do the drifts $(f_i)$ and the diffusions components $(\sigma_i)$ satisfy the same assumptions in Theorem 1?
- In Equations (16) and (17),  the vectors  $\boldsymbol{X}(t_i)$ and $\hat{\boldsymbol{X}(}t_i)$ are $d$-dimensional then the squared  $(\boldsymbol{X}(t_i) -\hat{\boldsymbol{X}(}t_i))^2$ must be a squared-$L_2$ norm.

- For the rotated  $W_2$, I couldn't understand the intuition behind this application of rotations. I think there is an "embedding"-like of the origin components $X_1$ and $X_2$ of the SDE, hence it seems that we have changed the origin SDE, by changing its drift and diffusions components. So, the rotated $W_2$ is it an upper or a lower bound for the origin one?
- Does the rotated squared $W_2$ verify the same upper and lower bounds as in Theroems 1 and 2?
- In the definition of the rotated Wasserstein, does $m$ the number of rotations? If yes, so in Tables 6 and 7 $N_{rotate}$ should be replaced by $m$.
- According to Figure 4, I would prefer the reconstruction of $f$ and $\sigma$ using Eq. 16 and Eq. 17 than the rotated $W_2$. Indeed, according to their definitions on Page 17,  they are acting on the empirical observations of the process in the time partition $(t_i)$. However, the rotated $W_2$ is acting on an embedding of the origin process, which might induce some loss of data information.

- Tables 6 and 7 could be more readable if they were represented as plots.

**Questions:**

### Minor comments
- Page 2: "regularized W distance", add the reference (Cuturi 2013, NIPS'13).
- Page 2: "reconstructingmultidimensional"  -->  reconstructing multidimensional.
- Page 4: "MSE-based": the acronym MSE is not defined.
- Page 6: "torchsde package (Lie tal, 2020) in Python" --> "torchsde Python package (Lie tal, 2020).
- Page 7: "Cox-Ingersoll-Ross" --> "Cox-Ingersoll-Ross (CIR)".
- Page 8: "Python Optimal Transport package": add its reference Flamary et al. JMLR'21.
- Page 8, Caption Figure 4: "the vectors $f(X_1, X_2) \hat f(X_1, X_2)$" --> add a comma $f(X_1, X_2), \hat f(X_1, X_2)$.
- Page 12: Equation (25): there is not a norm, it is only an absolute value.
- Page 14: "where $||X||^2$: $X$ would be in bold.
- Page 14: $| \cdot |$ is the $l^2$ norm: this is a little bit confusing because this notation is used before for the absolute value.
- Page 14: In Equation (34), $\ell$ is not defined.
- Page 16: "two vectors $(x_1(t_i), X_2(t_i)$: missing closed parenthesis.
- Page 21: Caption of Table 7: "This indicates the correlation .... in the drift term", I think the correlation in the drift is more difficult to distinguish than the correlation in the diffusion because the relative error in $\sigma$ gets much smaller.

---

> ### Author Response · Authors · 2023-11-16
> **Rebuttal responses to Reviewer NdWD part I**
>
> Thank you for your careful reading and valuable suggestions.  We have
> revised our manuscript based on your comments.
>
> To summarize, to the best of our knowledge, there are very few
> published methods for general SDE reconstruction.  We propose the
> first general framework that can be readily used to reconstruct
> general SDEs based on the Wasserstein distance. It provides a simple
> and efficient (compared to recent Wasserstein generative adversarial
> method and MMD loss method) way to reconstruct SDEs.  We applied our
> new Wasserstein-distance-based SDE reconstruction method noisy
> trajectories that arise in many biological and biophysical
> settings. We show some of our preliminary results here
> https://drive.google.com/file/d/1HW5CBjx7g37DFx3s3A-sNDK0aoG6zqbT/view?usp=drive_link.
> We will submit
> more extensive results on arXiv soon.
>
> Given the novelty and potentially broad applicability of our method
> towards uncertainty quantification and analyzing time series data, we
> hope you will kindly consider reevaluating the revised manuscript and
> update your score in light of our responses:
>
> W1. We have added a sentence in Theorem 1 to indicate that it
> applies to the 1D SDE case.
>
> W2. We concur that presenting $d$-dimensional version of Theorem 1
>   is beneficial.  As we
> have shown in the remark, Theorem 1 can be straightforwardly extended
> to $d$-dimensional SDEs in the trivial case where $f_i, \sigma_i$
> depend only on $X_i$ (the $i^{\text{th}}$ component of the vector
> ${\boldsymbol X}$). We are currently working on generalizing Theorem 1 to
> coupled multidimensional SDEs and have revised our remark accordingly
> to state that multivariate cases is quite nontrivial.
>
> Even though we have not yet been able to obtain theoretical results
> that generalize Theorem 1 to multidimensional SDEs, Eq. (17)
> nonetheless works well as a loss function for reconstructing
> multidimensional SDEs, as is shown in our Example 4 and the additional
> examples we provide.
>
> In summary, while we await proofs of the generalization of Theorem 1
> to multidimensional SDEs, we have highlighted the practical success of
> our approach in reconstructing multidimensional stochastic systems. We
> remain committed to sharing any theoretical advancements in subsequent
> work, ensuring transparency and clarity in our contributions.
>
> W3. If each $f_i, \sigma_i$ depend only on $X_i, t$ and satisfy the conditions in
> Theorem 1, then Theorem 1 can be generalized to multivariate cases. We
> are still working on extending Theorem 1 to interacting
> multidimensional SDEs, which appears quite nontrivial.
>
> W4. We have revised
> the $\ell^2$ vector norm notation $|\cdot|^2$ to $|\cdot|^2_2$ to avoid confusion.
>
> W5.  We
>   appreciate your inquiry regarding the intuition behind the
>   application of rotations in the rotated $W_2$ method. The rotated
>   $W_2$ serves as a lower bound of the true $W_2$ distance. In higher
>   dimensions, an analytical evaluation of the $W_2$ distances is
>   challenging. Our approach approximates the $W_2$ distance in two
>   ways: the decoupled $W_2$ and the rotated $W_2$.
>
>   The primary method, which we refer to as the decoupled $W_2$
> distance, involves projecting the 2D distribution in two orthogonal
> directions and summing the $W_2$ distances between $1 \mathrm{D}$
> projections of the true and reconstructed data. This method yields a
> lower bound of the true $W_2$ distance and is exact (equal to the true
> $W_2$) if the 1D projections $X_1$ and $X_2$ are mutually independent.
>
> However, this approach may not effectively handle skewed (correlated)
> distributions. We address this by applying a suitable rotation, as
> illustrated in the figure https://drive.google.com/file/d/1wJdYAzZQhNEEOsQ-80Ne1GvWsuxxX0D-/view?usp=drive_link.
>
> Practically, the
> optimal rotation matrix is unknown, necessitating an average over
> various rotations. Since each rotation is isometric, rotation
> does not change the true $W_2$ distance, and thus each decoupled
> $W_2$ distance post-rotation remains a lower bound of the true $W_2$
> distance. Therefore, the averaged $W_2$ distance after rotation also
> constitutes a lower bound of the true $W_2$ distance.
>
> We discuss the rotated $W_2$ distance in more detail in Appendix D of the revised manuscript
> as well as in the response to Comment 8 below.
>
>
> W6. The rotated squared $W_2$ is a lower bound
> of the true $W_2$ distance and aims to approximate the true $W_2$
> distance for the empirical distributions on the time partition
> $\left(t_i\right)$.  Consequently, upper bounds for $W_2$ distances
> associated with the time partition $\left(t_i\right)$ hold for the
> rotated squared $W_2$ distance as well.  We have further clarified the
> definition of the rotated squared $W_2$ loss in Appendix D of the
> revised manuscript.
>
> W7. Yes, $m$ is the number of rotations. We
> have revised the notation in Tables 6 and 7 (now Fig. S1 and Fig. S2)
> accordingly.

---

> ### Author Response · Authors · 2023-11-16
> **Rebuttal responses to Reviewer NdWD part II**
>
> W8. To clarify, the rotation itself does not introduce additional
> loss of information, as the rotation is an isometric linear
> transformation. From an abstract perspective, the true
> high-dimensional dynamics is not changed by a change of coordinates.
> From another perspective, the original SDE of $\boldsymbol{X}_t$ and
> the rotated SDE of $\boldsymbol{Y}_t \coloneqq
> \boldsymbol{R}\boldsymbol{X}_t$, where $\boldsymbol{R}$ is a rotation
> matrix, are related by
> \begin{equation}
> \mathrm{d} {\boldsymbol{Y}} = \boldsymbol{R}
> \boldsymbol{f}\big(\boldsymbol{R}^{-1} (\boldsymbol{Y})\big)
> \mathrm{d} t + \boldsymbol{R} \boldsymbol{\sigma} \big(\boldsymbol{R}^{-1}
> (\boldsymbol{Y})\big)\mathrm{d} \boldsymbol{W}_t.
> \end{equation}
>  Once knowing the
> rotated SDE, we can recover the original SDE by applying the inverse
> transform $\boldsymbol{R}^{-1}$.
> Despite this, averaging over different rotations and projecting in
> different directions can lead to some loss of information.
>
> Lastly, we plug the decoupled and rotated $W_2$ distances into Eq. 17
> as approximations to the true $W_2$ distance at each time point.
> While these methods provide analytical approximations to
> the true $W_2$ distance, our manuscript also explores numerical
> optimization procedures for a numerical approximation of the $W_2$
> distance. The purpose is to compare the performance and computational
> costs of these methods. We discuss these
> aspects in further detail in Example 4 of the revised manuscript.
>
>
> W9. We have replaced Tables 6 and 7 with Figs. S1
> and S2 in the revised manuscript.
>
> $\textrm{\textbf{Minor comments:}}$
>
> We really appreciate your careful reading and we have made revisions
> based on your minor comments and cited the references you
> suggested. Here are our responses to some of your questions in the
> minor comments.
>
>
>
> Q11. In Equation (34), $\ell$ is was added by mistake.
>   We have removed it in the revised manuscript.
>
> Q13. Thank you for your observation regarding the correlation in the drift
> term. We apologize for any confusion caused by our initial
> explanation. In the revised manuscript, we have clarified the
> comparison between the two scenarios outlined in Tables 6 and 7 (now
> Fig. S1 and S2). In Table 6 (now Fig. S1), the ground truth SDE is
> defined by Eq. 22, where the drift terms have no correlation across
> different components. In Table 7 (now Fig. S2), the ground truth SDE
> is defined by Eq. 48, where the drift terms are correlated across
> different components. In both cases, the diffusion terms are
> correlated across different components.
>
> Looking at the relative errors in drift terms $\boldsymbol{f}$, they all
> hover around 0.1 for both cases and all values of rotation number $m$.
> Comparison between Tables 6 and 7 (now Fig. S1 and S2) shows that
> correlation in drift terms \textit{does not} significantly affect
> the reconstruction of drift terms.
>
> By contrast, increasing the rotation number $m$ from 1 to 10
> significantly improves the reconstruction of diffusion terms
> $\boldsymbol{\sigma}$ when they are correlated across different
> components (Table 7, now Fig. S2).
>
> In conclusion, the rotated $W_2$
> distance ($m \geq 2$) performs better than the decoupled $W_2$
> distance ($m = 1$) in reconstructing correlated diffusion terms, while
> it does not significantly contribute to the reconstruction of drift
> terms. On the other hand, the existence of correlation in diffusion terms
> confounds the reconstruction by the decoupled $W_2$ distance, while correlation
> in drift terms does not. This suggests that the existence of correlation in diffusion terms is indeed more
> difficult to distinguish than the existence of correlation in drift terms in our example. In reconstructing general multivariate SDEs,
> it remains to explore whether the correlation in the drift is more difficult than the correlation in the diffusion
> to distinguish or not.
>
> We have revised the paragraph following Table 7 (now Fig. S2) to
> clarify this point.
>
> We hope that our responses satisfactorily address your questions and
> concerns and that you will better appreciate the new
> Wasserstein-distance-based SDE reconstruction method we have
> developed.

---

> ### Author Response · Authors · 2023-11-21
>
> Dear Reviewer NdWD,
>
> We are extremely grateful for the time and expertise you dedicated to reviewing our paper. Your suggestions and feedback greatly helped us improve our work. We deeply value your support and guidance in this process.
> In light of the changes we have made in response to your comments, we would be immensely appreciative if you could take a moment to review our revisions. We would be greatly appreciating it if our responses have satisfactorily addressed your concerns and warrant your reevaluation.
> Thank you once again for your invaluable contribution and for sharing your expertise with us!
>
> With sincere gratitude,
>
> Authors

---

### Official Review · Reviewer_pb7V · 2023-11-03

**Soundness:** 3 good
**Presentation:** 2 fair
**Contribution:** 2 fair
**Rating:** 6
**Confidence:** 4

**Summary:**

Given SDE of form (2), expressions for optimal transportation between the two such process is provided. An upper bound on the objective is shown in theorem1, which guarantees that when both deterministic and stochastic functions in the SDE are close, then the objective is close to zero. Then a time-discretised version (16) is presented, which by theorem2 convergence to the true objective as time-step becomes infinitesimally small.

The optimal objective is then employed as a loss to learn/estimate an SDE. Simulations on simple SDEs are presented.

**Strengths:**

1. SDE estimation using optimal transport seems interesting. I feel this is not so well-understood from a learning perspective and perhaps deserves more attention.

**Weaknesses:**

1. The presentation can be improved to make it easily readable. Currently, I found it hard to understand what is happening. Brevity and notation compound the difficulty.

2. Since the bound (7) is not elegant, perhaps it could be simplified appropriately to preserve the point about convergence.

3. Some training set details for simulations seem missing like what was the step-size, time-horizon, number of samples etc ?

4. There seem to be prior works which perhaps present a more comprehensive analysis of SDE wrt. optimal transport e.g., [1*], [2*], [3*], [4*]. It would be insightful if these are discussed in detailed in related work and perhaps appropriately compared with. I feel this is a major weakness of the work.

[1*] https://arxiv.org/pdf/1902.08567.pdf
[2*] https://arxiv.org/pdf/1603.05484.pdf
[3*] https://arxiv.org/abs/1209.0576
[4*] https://www.jmlr.org/papers/volume22/21-0453/21-0453.pdf

**Questions:**

1. What do the measures \mu,\hat{\mu} represent? It seems they are measures corresponding to X(t) and \hat{X}(t) . If so, should not there be a \mu(t), \hat(\mu}(t) ? In which case, they cannot be used in the LHS of 5 as the RHS is integrated over time. I think the RHS is a generalization of optimal transport to SDEs, but the optimal objective need not be a wasserstein distance (LHS). Am I correct ?

2. why are different sets of baselines used in the various simulation examples ?

---

> ### Author Response · Authors · 2023-11-16
> **Rebuttal responses to Reviewer pb7V part I**
>
> Thank you for your valuable comments.  We have revised our manuscript
> based on your suggestions.
>
> To summarize, to the best of our
> knowledge, there are very few published methods for general SDE
> reconstruction.  Addressing this gap, we introduce a novel framework
> that uses the Wasserstein distance and that exhibits efficiency in
> reconstructing general SDEs across a number of applications. Our
> approach offers a straightforward and efficient alternative relative
> to recent methods such as Wasserstein generative adversarial and MMD
> loss methods for SDE reconstruction.
>
> To demonstrate its practical applicability we have used our new
> Wasserstein-distance-based SDE reconstruction method to reconstruct
> SDEs from noisy trajectories arising in some biological and biophysics
> settings. We present some of our preliminary results here https://drive.google.com/file/d/1HW5CBjx7g37DFx3s3A-sNDK0aoG6zqbT/view?usp=drive_link, with
> more extensive results posted on arXiv soon. Given the novelty and
> potentially broad applicability of our method towards uncertainty
> quantification and analyzing time series data, we hope you will
> kindly reevaluate the revised manuscript
> and score it in light of our responses:
>
> W1. We have taken your comments into
> consideration and made improvements to enhance the readability of our
> paper. Specifically, we have reorganized subsection 1.2, focusing on
> clearly presenting the structure of our paper and emphasizing the
> novelty of our contributions. We believe that the roles of Theorem 1
> and Theorem 2 in the analysis and development of our
> Wasserstein-distance SDE reconstruction method, along with key aspects
> of our numerical experiments, are now more clearly introduced. We hope
> that our revisions could satisfactorily address your concern.
>
> W2.  In Appendix A, we provide a detailed proof of Theorem 1, outlining each
> intermediate step and the Cauchy inequalities employed to derive the
> error bound. While the current form of Eq. (7) may not be the most
> elegant, it effectively communicates how the Wasserstein distance
> $W_2(\mu, \hat{\mu})$ is bounded by multiples of the differences in
> the drift and diffusion terms, $f-\hat{f}$ and $\sigma-\hat{\sigma}$,
> between two SDEs. Eq. (7) serves to illustrate the necessity of
> minimizing $W_2(\mu, \hat{\mu})$ for SDE reconstruction, aiming to
> reduce the errors $f-\hat{f}$ and $\sigma-\hat{\sigma}$ between the
> ground truth and reconstructed SDEs.
>
> We acknowledge the potential for further refinement by imposing
> tighter inequalities in Appendix A to enhance the error bound.  What
> is more interesting and nontrivial is to extend the error bound
> Eq. (7) to solutions between two multidimensional SDEs.  We are
> currently working on multidimensional SDEs but the analysis is
> nontrivial, as described in Remark 1.
>
> W3.  At the beginning of Section 3, we have written that training
> details for all examples such as the step-size, the number of samples,
> the learning rate, etc, are given in Appendix E. The time-horizon is
> given separately in each example (Eqs. (18), (20), (21), (23)).

---

> ### Author Response · Authors · 2023-11-16
> **Rebuttal responses to Reviewer pb7V part II**
>
> W4. We appreciate your providing these
> references.  We were aware of some of these and had cited Reference
> [2] from your list in our manuscript. However, these previous works do
> not directly address the specific task of approximating an SDE by
> another SDE or SDE reconstruction. Moreover, the references you
> provided primarily focus on the Wasserstein distance between solutions
> to the same SDE with different initial conditions or other specific
> scenarios, such as stochastic contraction and numerical solutions;
> they do not directly address the SDE reconstruction problems that our
> proposed method aims to solve. We summarize them here:
>
>  (1) Reference [1] analyzes the Wasserstein distance between
>   solutions to the same SDE with different initial conditions, which
>   is called stochastic contraction. It is not directly applicable to
>   approximating an SDE by another SDE.
>
>  (2) Reference [2] studies the Wasserstein distance between solutions
>   to the same L\'evy process \& potential jumps with different initial
>   conditions as well as coupling. It is not directly applicable to
>   approximating an SDE by another SDE or SDE reconstruction problems
>   either.
>
>  (3) Reference [3] investigates the Wasserstein distance between the
>   solution to an SDE and its Euler scheme approximation.  Yet, the results
>   cannot be directly applied
>   for SDE reconstruction, which requires analyzing the Wasserstein distance
>   between solutions to two SDEs.
>
>  (4) Reference [4] studies the Wasserstein distance between the
>   solution to an ergodic SDE and its numerical solution. It mainly
>   discusses how to appropriately obtain the numerical solutions to
>   SDEs of some specific form.  This paper does not mention how to
>   reconstruction SDEs.  Yet, the Langevin equations it analyzes could
>   be helpful if we wish to apply the Wasserstein distance to construct
>   processes other then SDEs. We have cited this paper in our revised
>   literature review.
>
> In our Appendix F, we also carried out a numerical experiment showing
> how the uncertainty in the initial condition affects the
> reconstruction of an SDE. Of course, it would be helpful to carry out
> more theoretical analysis on this, as was done in the references you
> kindly provided to us.
>
> As you and Reviewer 4 suggest, there seems to be no simple method
> so far that exploits the Wasserstein distance for SDE
> reconstruction. Thus, our method is not only new but it also appears
> to be quite efficient in applications we explored. The most recent and
> related work that directly tackles general SDE reconstruction or using
> one SDE to approximate another unknown SDE are the WGAN-SDE method and
> its extensions (Neural SDEs as infinite-dimensional GANs) and the MMD
> method (Generative moment matching networks). We compared our
> Wasserstein-2-distance-based SDE reconstruction method with two
> methods in Example 3 and found our method was more
> efficient and accurate. Furthermore, minimization of the Wasserstein-distance loss
> function (Eqs. 16, 17) used in our method is much easier to implement than
> the aforementioned WGAN and MMD methods.
>
> Q1. The measures $\mu, \hat{\mu}$ are two probability measures associated with
> the entire infinite dimensional temporal trajectories $\boldsymbol{X}(t), t\in[0, T]$ and $\hat{\boldsymbol{X}}(t),  t\in[0, T]$
> rather than specific probabilities for $\boldsymbol{X}(t), \hat{\boldsymbol{X}}(t)$ at a fixed time point $t$. Therefore, we would not
> use the notation $\mu(t), \hat{\mu}(t)$.
> The distance between two trajectories is thus defined as the temporal integration $\lVert\boldsymbol{X}\rVert\coloneqq
> \big(\int_0^T \sum_{i=1}^d |X_i(t)|^2\text{d} t\big)^{\frac{1}{2}}$. In this context, we can define
> the Wasserstein-2 distance between the two measures $W_2(\mu, \hat{\mu})$ as Eq. (5).
>
> Q2. In Examples 1 and 2, we compare our method with several classical uncertainty quantification (UQ) methods.
> We find that the Wasserstein distance could indeed be better in reconstructing SDEs' trajectories and drift as well as diffusion terms than those classical UQ loss functions in statistics.
> In Examples 3 and 4, for simplicity, we do not compare with those classical statistical UQ methods again. Instead, we focus on comparing with two most recent machine-learning-based SDE reconstruction methods (WGAN and MMD). We find that our method could also outperform these two methods in SDE reconstruction.
>
> We hope that our responses satisfactorily address your questions and
> concerns and that you will better appreciate the new
> Wasserstein-distance-based SDE reconstruction method we have
> developed.

---

> ### Author Response · Authors · 2023-11-21
>
> Dear Reviewer pb7V,
>
> We are grateful for the time and expertise you dedicated to reviewing our paper. Your insightful feedback has been instrumental in guiding us to improve our work. We deeply value your support and guidance in this process.
> In light of the changes and clarifications we have made in response to your comments, we would be immensely appreciative if you could take a moment to review our revisions. Your confirmation on whether our responses have adequately addressed your concerns would be greatly valued.
> Thank you once again for your invaluable contribution and for sharing your expertise with us! We would greatly appreciate it if our revisions and changes warrant your reevaluation of our work.
>
> With sincere gratitude,
>
> Authors

---

> > ### Comment · Reviewer_pb7V · 2023-11-22
> >
> > I thank the authors for their detailed response. Since my concerns are addressed, I am increasing the score.

---

> > > ### Author Response · Authors · 2023-11-22
> > >
> > > Dear Reviewer pb7V,
> > >
> > > We really appreciate your time in reading our responses and your reevaluation!
> > >
> > > Extracting SDEs from noisy time series data has been an important topic in uncertainty quantification for biological and physical applications. However, there exist very few methods for SDE reconstruction. Using the W2 distance as a loss function to reconstruct SDEs is quite easy to implement and easy to understand. Furthermore, it outperforms traditional methods and recent machine learning methods we have known so far in both efficiency and accuracy. Thus, we believe that our proposed simple W2-distance-based loss SDE reconstruction method could become a potential benchmark method in this specific SDE reconstruction field. We would greatly appreciate it if you found our work interesting and potentially useful.
> > >
> > > Sincerely,
> > >
> > > Authors

---

### Comment · Reviewer_NdWD · 2023-11-21

I thank the authors for their detailed answers to my concerns, in particular for clarifying the rotated 2-Wasserstein. I decided to raise my score to 6. Furthermore, I think the reconstruction of the 5-dimensional SDE model of circadian clocks, given in the rebuttal, highlights the proposed approach in a real data application. I encourage the authors to add this and the figure for the explanation of the rotation (given also in the rebuttal) to the appendices.

---

> ### Author Response · Authors · 2023-11-22
> **Thanks for your suggestions! & Revisions based on your comments**
>
> Dear Reviewer NdWD,
>
> We really appreciate your feedback and your reevaluation!
>
> We are grateful for your constructive suggestions. Based on your suggestions, we have added the figure for illustrating the rotated W2 distance in Appendix D as well as added a new appendix section (Appendix K) detailing the reconstruction of the 5D circadian SDE model. The 5D circadian SDE model is still our ongoing work and we will post more details of it along with the application of our novel W2 distance SDE reconstruction method into other biological problems on arXiv soon. Please see our updated manuscript for the revisions.
>
> Extracting SDEs from noisy time series data has been an important topic in uncertainty quantification for biological and physical applications. However, there exist very few methods for SDE reconstruction. Using the W2 distance as a loss function to reconstruct SDEs is quite easy to implement and easy to understand. Furthermore, it outperforms traditional methods and machine learning methods we have known so far (the WGAN-SDE and the MMD method) in both efficiency and accuracy. Thus, we believe that our proposed simple W2-distance-based loss functions for reconstructing SDEs could become a potential benchmark method in this specific SDE reconstruction field. We would greatly appreciate it if you found our work interesting and potentially useful.
>
> Sincerely,
>
> Authors

---

### Meta-Review · Area_Chair_dCKw · 2023-12-07

**Metareview:**

The authors derive an upper of Wasserstein w.r.t. the discrepancies of deterministic, stochastic components ($f, \sigma$) between two stochastic difference equations (SDEs). The authors propose to use Wasserstein loss to reconstructing SDE from noisy data. The authors illustrate the advantages of the proposed methods over other baseline losses.

Excluding the short comment of Reviewer 2Kjt, other reviewers think that the proposed approach is interesting. Additionally, the reviewers have concerns about the theoretical results about using Wasserstein as a loss to reconstruct SDE (e.g., minimizing Wasserstein loss does not lead to close the gap between discrepancies of deterministic, stochastic components between two SDEs; and approximation of Wasserstein loss).

The empirical advantages of the proposed methods are the plus point. To my knowledge, the theoretical results do not suffice to support the usage of Wasserstein loss for SDE reconstruction yet. I encourage the authors to elaborate the raised issues on theoretical results rigorously to improve the submission.

**Justification For Why Not Higher Score:**

I lean on the negative side for the decision since the major point is not addressed yet (i.e., the proposed theoretical results do not suffice to support the usage of Wasserstein loss for SDE reconstruction yet although the authors draw some related connection.)

**Justification For Why Not Lower Score:**

N/A

---

### Decision · Program_Chairs · 2024-01-16

Reject